# SSRP1-mediated histone H1 eviction promotes replication origin assembly and accelerated development

Lucia Falbo [1], Erica Raspelli[1], Francesco Romeo[1], Simona Fiorani[2], Federica Pezzimenti[1], Francesca Casagrande[1], Ilaria Costa[1], Dario Parazzoli [1] & Vincenzo Costanzo[1,3 ✉]

In several metazoans, the number of active replication origins in embryonic nuclei is higher than in somatic ones, ensuring rapid genome duplication during synchronous embryonic cell divisions. High replication origin density can be restored by somatic nuclear reprogramming. However, mechanisms underlying high replication origin density formation coupled to rapid cell cycles are poorly understood. Here, using *Xenopus laevis*, we show that SSRP1 stimulates replication origin assembly on somatic chromatin by promoting eviction of histone H1 through its N-terminal domain. Histone H1 removal derepresses ORC and MCM chromatin binding, allowing efficient replication origin assembly. SSRP1 protein decays at mid-blastula transition (MBT) when asynchronous somatic cell cycles start. Increasing levels of SSRP1 delay MBT and, surprisingly, accelerate post-MBT cell cycle speed and embryo development. These findings identify a major epigenetic mechanism regulating DNA replication and directly linking replication origin assembly, cell cycle duration and embryo development in vertebrates.

[1] IFOM, The FIRC Institute of Molecular Oncology, Milan, Italy. [2] Clare Hall Laboratories, London Research Institute, Cancer Research UK, South Mimms, UK. [3] Department of Oncology and Haematology-Oncology, University of Milan, Milan, Italy. ✉email: Vincenzo.Costanzo@ifom.eu

Eukaryotic replication origins are assembled by the orderly recruitment of protein factors to initiation sites. The origin recognition complex (ORC) binds to replication origins and, together with Cdt1 and Cdc6, promotes the pre-replication complex (pre-RC) assembly by orchestrating the loading of the MCM helicase complex. The MCM complex remains inactive until Dbf4-dependent kinase (DDK) and cyclin-dependent kinases (CDKs) trigger the recruitment of initiation factors Cdc45 and GINS forming the CMG complex, which together with DNA polymerases duplicates the DNA[1]. Factors such as chromatin configuration, DNA sequence and shape and other proteins including histone acetylation enzymes, chromatin remodelers and histone chaperones affect pre-RC assembly. However, their precise roles are poorly understood[2].

In *Xenopus laevis* early embryos and egg extracts DNA replication starts from replication origins separated by an average inter-origin distance of 9–12 kb (refs. [3,4]). This replication origin density persists for the first 12 synchronous cell cycles, up to the MBT, when cell division becomes asynchronous, transcription is activated and slower and asynchronous somatic cell cycles start[5]. Somatic cells, instead, replicate DNA from fewer replication origins. Highly differentiated cells such erythrocytes assemble origins at an average inter-origin distance of 100–120 kb (ref. [6]).

Titration of replication initiation factors Cdc45, Drf1, TopBP1, and Treslin has been in part linked to the lower replication origin density at MBT[7]. However, embryonic replication density is not restored when somatic nuclei are incubated in interphase egg extracts despite the excess of replication initiation factors[6], indicating the existence of additional mechanisms that prevent replication origin assembly on somatic nuclei. Chromatin configuration could contribute to the decreased number of origins on post-MBT somatic nuclei by suppressing the chromatin binding of replication origin components. Consistent with this, decreased replication origin density correlates with decreased levels of ORC complex loaded onto somatic DNA, indicating that chromatin-bound factors upstream ORC may regulate DNA ORC binding and distribution[6].

Intriguingly, embryonic replication origin density can be restored on somatic nuclei by incubation in intact unfertilized eggs or their mitotic arrested extracts[6,8,9]. This process is accompanied by active chromatin remodeling and removal of somatic chromatin-bound proteins such as transcription factors. Replication origin re-configuration is thought to be essential for nuclear reprogramming obtained through somatic cell nuclear transfer[6,9,10].

Somatic nuclei contain high amounts of histone H1, which contribute to chromatin compaction[11] and could restrain replication origin assembly. Here, we show that SSRP1, through its N-Terminal region, promotes histone H1 removal from somatic chromatin, licensing replication origins assembly on somatic nuclei in egg extracts. SSRP1 together with SPT16 forms the FACT complex, a major chromatin remodeller[12–14]. Critically, we show that SSRP1 protein levels, decrease at MBT when somatic DNA replication and asynchronous cell cycles start. SSRP1 overexpression can significantly delay the onset of MBT and somatic cycles. Strikingly, SSRP1 sets the speed of post-MBT development as embryos exposed to higher SSRP1 protein levels develop at a significant faster pace. This is likely due to enhanced replication origin assembly, which increases the speed of genome duplication and cell cycle. These findings indicate that chromatin configuration and replication origin assembly are directly linked to the control of somatic cell cycle duration in vertebrate development. Considering that high levels of SSRP1 and the FACT complex have been shown to drive tumor growth[15] and that recent work highlighted a key role for the linker histone H1.0 in suppressing cancer cell proliferation and self-renewal[16,17] our results set the stage to better understand this central epigenetic regulation in DNA metabolism and cell cycle.

## Results

### Isolation of SSRP1 as somatic nuclei replication activator.

Somatic nuclei (~4000 nuclei/µl) derived from *Xenopus* erythrocytes do not replicate efficiently in interphase egg extract in which replication factors are not limiting. However, prolonged pre-incubation of somatic nuclei in cytostatic factor arrested (CSF) mitotic extract derived from unfertilized eggs allows their efficient replication in interphase extract[6] (Fig. 1a). Collectively, these observations suggest that inhibitory factors on somatic chromatin prevent DNA replication and that these are removed in unfertilized mitotic eggs and extracts. Unexpectedly, by titrating somatic nuclei we found that a low number of nuclei could also be efficiently replicated in interphase egg extract similar to sperm nuclei, the number of which did not significantly affect replication efficiency[18] (Fig. 1b). These observations indicated that interphase extracts contain factors present in limiting amounts able to remove inhibitory constraints preventing efficient somatic nuclei replication. To identify these factors, we fractionated and concentrated interphase egg extract using polyethylene glycol (PEG). Fractions precipitated with increasing concentrations of PEG were pre-incubated with somatic nuclei, which were then transferred to interphase egg extracts (Fig. 1c, d) to assay their ability to stimulate DNA replication. The fraction with the highest activity was recovered from the 9% PEG pellet, which contained several proteins (Supplementary Fig. 1a). To identify the factor responsible for this effect the active fraction was separated by column chromatography to isolate DNA binding proteins able to stimulate somatic nuclei replication (Fig 1e). The active PEG and column fractions were analysed by gel free multi-dimensional protein identification tandem mass spectrometry (MS)[19]. We found a high number of SSRP1 peptides together with lower amount of SPT16 peptides in the fraction with the highest stimulatory specific activity (Supplementary Data 1). As somatic chromatin configuration was possibly responsible for the inhibition of somatic nuclei replication we tested the ability of SSRP1 and SPT16 to stimulate somatic DNA replication. To this end we produced recombinant human Flag-SSRP1 and 6xHis-SPT16 proteins, which are highly similar to the *Xenopus* orthologs and could be easily expressed in soluble form (Supplementary Fig. 1b). We found that recombinant SSRP1 alone and in complex with SPT16 pre-incubated with somatic nuclei was able to stimulate their replication in interphase extract without prior incubation in CSF-arrested mitotic egg extracts (Fig. 2a, b). The stimulatory effect was linearly correlated to the amount of SSRP1 protein and was already detectable at 50 ng/µl, corresponding to 545 nM, 2.8-fold higher than endogenous SSRP1 concentration, which was estimated to be ~190 nM (ref. [20]). Noticeably, SPT16 alone did not stimulate DNA replication, suggesting that SSRP1 was sufficient to produce this effect (Fig. 2a).

To test whether SSRP1 was acting on the general DNA replication machinery we incubated SSRP1 with sperm nuclei. In this case SSRP1 was unable to stimulate DNA replication when sperm nuclei were used as template, indicating that SSRP1 was only able to promote replication of somatic nuclei (Fig. 2c), excluding that SSRP1 was directly affecting the efficiency of the DNA replication machinery.

### SSRP1 stimulates DNA replication origin assembly.

To identify the molecular mechanism of SSRP1-dependent somatic DNA replication stimulation, we monitored replication origin distribution

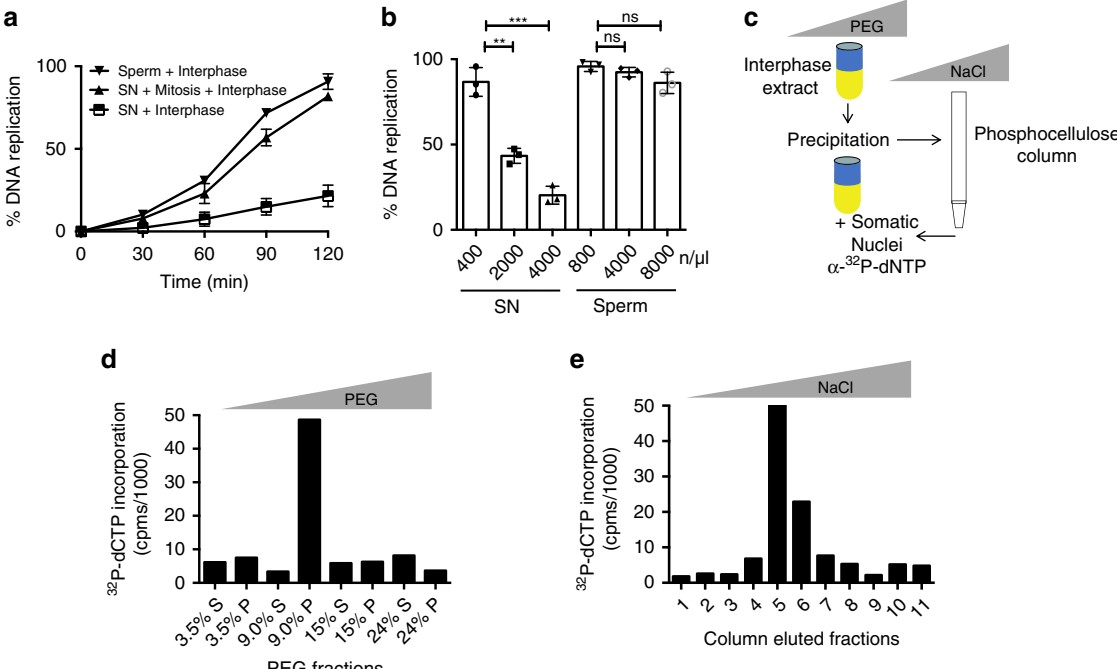

**Fig. 1 Isolation of SSRP1 as somatic nuclei replication stimulating factor. a** Graph showing DNA replication kinetics of sperm (8000 n/µl) or somatic nuclei (SN) (4000 n/µl). DNA was directly incubated in interphase egg extract (Interphase) or exposed for 30 min to CSF-arrested mitotic extract, which was then driven into interphase by 0.4 mM CaCl$_2$ addition (Mitosis + Interphase). DNA synthesis was quantified by measuring the percentage of α-$^{32}$P-dCTP incorporation relative to the input DNA for each condition. Each point represents the mean value ± standard error of the mean (SEM). $n = 3$ independent experiments; $p < 0.0001$ when comparing mean values for all points; Two-way anova. **b** Graph showing DNA replication of the indicated amounts of sperm and SN directly incubated in interphase egg extract for 120 min. DNA synthesis was quantified as in (**a**). Bars represent mean ± SEM. $n = 3$ independent experiments; **$p < 0.01$, ***$p < 0.001$ or ns when comparing mean values for the indicated samples; unpaired $t$ test. **c** Schematic representation of *Xenopus* egg extract fractionation. Extract was precipitated by centrifugation following incubation with increasing amount of polyethylene glycol (PEG). Either pellet (P) or supernatant (S) fractions collected at different PEG concentrations were assayed for their ability to stimulate replication when incubated with SN then transferred to interphase extracts in the presence of radiolabelled nucleotide. **d** Graph showing α-$^{32}$P-dCTP incorporation (counts per minute, cpms) in SN pre-incubated with the indicated PEG fractions for 30 min and then transferred to interphase egg extract for 120 min. **e** Graph showing α-$^{32}$P-dCTP incorporation in somatic nuclei pre-incubated with the fractions eluted from the column as described in (**c**) loaded with the active PEG fraction (9%P) and then transferred to interphase egg extract for 120 min.

by DNA combing. Somatic nuclei were incubated in interphase extract in the presence of DNA modified precursor digoxygenin-11-dUTP (Dig-U), which labels initiation sites[21] (See Methods). The center-to-center distance between adjacent Dig-U tracks (or eyes, which were considered to be the product of one replication fork bubble) was considered as inter origin distance (IOD). IODs were measured for each treatment and plotted on a graph (Fig. 2d), revealing that somatic nuclei assembled a low number of origins at IODs of 90–110 Kb on average (Fig. 2e) and at a density of one origin every ~100 Kb. Strikingly, we observed a significant increase in the number of replication origins in somatic nuclei pre-incubated with SSRP1 and then added to interphase egg extracts. SSRP1 stimulated the formation of origins, which assembled at IODs of 10–15 Kb (Fig. 2e) and at a density of 6–8 origins every 100 Kb. These values were similar to the ones obtained by pre-incubation of somatic nuclei in CSF arrest mitotic egg extract (Fig. 2d, e). Consistent with these results the amount of ORC and MCM complexes bound to chromatin was significantly increased by SSRP1 (Fig. 2f, g). Thus, pre-incubation of somatic nuclei with recombinant SSRP1 was sufficient to increase the density of active replication origins mimicking the one typically present during embryonic DNA replication. This process was mediated by the stimulation of ORC and MCM loading onto somatic chromatin. The stimulatory effect was already evident at early time points (Fig. 2f, 5 min lane), suggesting that SSRP1 acts on chromatin before or at the time of ORC binding. SSRP1-mediated increase of

replication origin density was independent of any transcriptional role possibly associated to the FACT complex as transcription is absent in *Xenopus* egg extract.

**SSRP1 averts histone H1-mediated origin assembly inhibition.**
Given these results we hypothesized that replication origin assembly on somatic chromatin could be prevented by an inhibitor that SSRP1 is able to remove. Among possible candidates we focused on somatic forms of histone H1, which can all reduce ORC binding and replication initiation events on sperm nuclei when overexpressed[22]. The histone H1 isoform present on erythrocyte chromatin is known as H1.0, henceforth referred as histone H1 (refs. [22,23]). We monitored histone H1 bound to chromatin incubated in interphase and CSF-arrested mitotic egg extracts. These experiments showed that histone H1 bound to somatic chromatin was removed in CSF-arrested but not interphase extracts (Fig. 3a). Strikingly, SSRP1 was able to promote significant removal of histone H1 when pre-incubated with somatic nuclei then added to interphase egg extract (Fig. 3b). The effect was specific for histone H1 as the chromatin binding of nucleosomal histones was unaffected (Fig. 3b). Noticeably, the levels and the migration pattern of endogenous SSRP1 protein in CSF-arrested mitotic extracts with high cyclin B levels were identical to the same extracts released in interphase by addition of CaCl$_2$ (ref. [6]), which induced cyclin B degradation, as expected

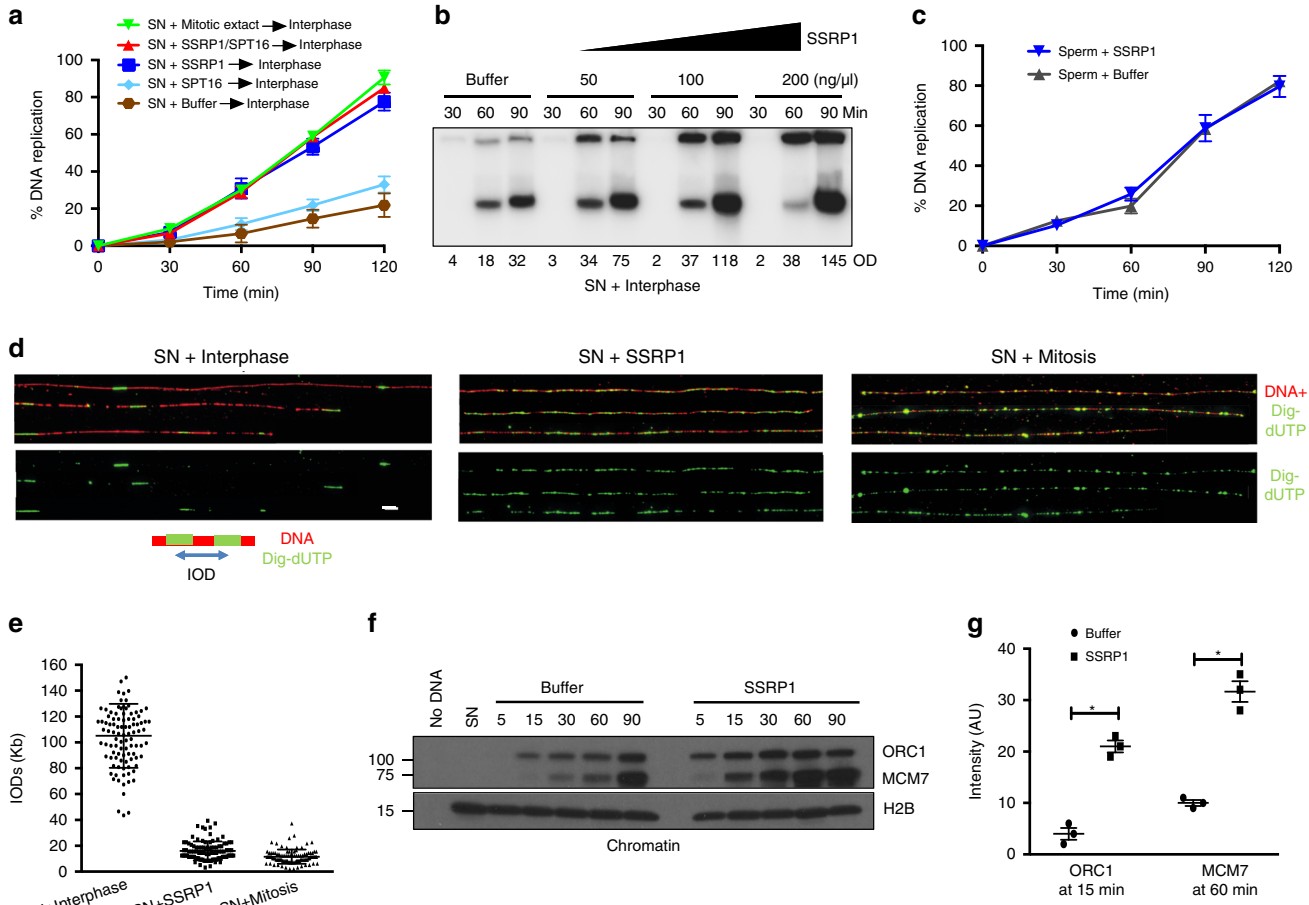

**Fig. 2 SSRP1-dependent stimulation of replication origin assembly. a** Graph showing replication time course of SN (4000 n/μl) replication. SN were pre-incubated for 30 min with buffer, 200 ng/μl recombinant SSRP1, SPT16 or both, as indicated, and transferred to interphase extracts (Interphase) or pre-incubated in CSF-arrested mitotic egg extract for 30 min (Mitotic) then driven into interphase by CaCl$_2$ addition. Each point represents the mean ± SEM. $n = 3$ independent experiments; $p < 0.0001$ when comparing mean values for all samples; Two-way anova. **b** Representative autoradiography of a DNA replication time course assay in interphase egg extract showing α-$^{32}$P-dCTP incorporation in SN. Optical density (OD) for each lane is indicated. **c** Graph showing replication time course of sperm nuclei (8000 n/μl) treated as indicated. Each point represents mean ± SEM. $n = 3$ independent experiments; $p =$ ns when comparing mean values for all samples; Two-way anova. **d** DNA fiber assay showing digoxigenin-dUTP (dig-dUTP) incorporation in SN incubated directly in interphase egg extract together with buffer (left), pre-incubated with 200 ng/μl recombinant SSRP1 (center) or pre-incubated for 30 min in CSF-arrested mitotic egg extract then driven into interphase by CaCl$_2$ addition (right). DNA fibers are in red. Dig-dUTP labeled tracts are in green. Bar = 10 Kb. **e** Graphs showing IODs indicated in Kb for each sample. The center-to-center distances between adjacent Dig-dUTP tracks were measured and plotted analyzing 50 fibers pooled from three independent experiments; IODs and mean values ± SEM are shown; $n = 100$ individual IODs; $p < 0.0001$; Two-way anova. **f** Chromatin binding showing somatic nuclei incubated in interphase extract in the presence of buffer or recombinant SSRP1. Chromatin was isolated at the indicated times after addition to egg extract and blotted using the indicated antibodies. No DNA indicates absence of nuclei. SN indicates somatic nuclei alone. Image shows typical result. **g** Graph showing relative quantification of signal intensity of ORC1 and MCM7 proteins loaded on chromatin at the indicated times normalized to histone H2B. Bars represent mean intensity ± SEM. $n = 3$ independent experiments; $p < 0.0005$ when comparing mean values for ORC1 and MCM7; $t$-test.

(Supplementary Fig. 2a), suggesting that the mechanism of CSF-arrested extract mediated removal of histone H1 does not rely on SSRP1. Prolonged exposure to high levels of mitosis promoting factor (MPF) present in CSF-arrested extract, which is known to phosphorylate histone H1 (refs. [24,25]), could instead contribute to histone H1 removal from somatic chromatin.

We then tested whether SSRP1 could directly counteract the inhibitory effect of histone H1 on replication origins assembly. To this end we tested the ability of purified histone H1 to inhibit DNA replication when incubated with sperm DNA. Consistent with previous work[22], H1 binding to sperm chromatin was able to prevent the assembly of replication origins as shown by inhibition of ORC and MCM binding to chromatin (Fig. 3c). This correlated with reduced replication origin number (~1–2/100 kb) placed at an average IOD of 90 Kb, as measured by DNA combing

(Supplementary Fig. 2b, c). Accordingly, α$^{32}$P-dCTP incorporation in DNA replication assays was strongly reduced by histone H1 (Fig. 3d). Significantly, pre-incubation of SSRP1 with sperm nuclei prevented histone H1-mediated inhibition of origin assembly and DNA replication (Fig. 3c, d). This correlated with increased replication origin number (~6–7/100 Kb) assembled at an average IOD of 12–15 Kb (Supplementary Fig. 2b, c). SSRP1 also counteracted chromatin compaction of sperm nuclei induced by histone H1 (Supplementary Fig. 2d, e). Overall, these experiments indicate that SSRP1 prevents histone H1-mediated inhibition of replication origin and promotes its eviction from somatic chromatin.

**SSRP1 N-terminus directly evicts histone H1 from chromatin.** To determine the mechanism of SSRP1-mediated removal of

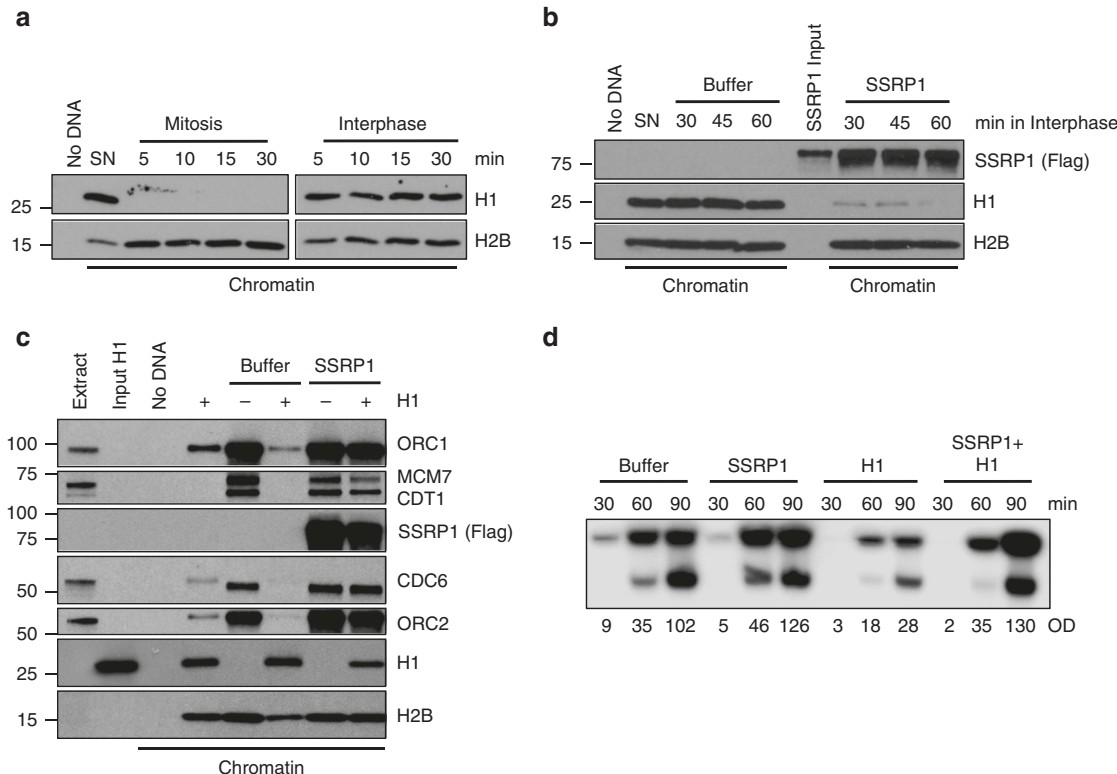

**Fig. 3 SSRP1 counteracts histone H1-dependent inhibition of origin assembly. a** Western blot (WB) of the indicated proteins bound to chromatin isolated from SN incubated with mitotic or interphase extracts for the indicated times. **b** WB of the indicated proteins bound to chromatin isolated from SN pre-incubated with 200 ng/μl recombinant SSRP1 or control buffer and transferred to interphase extracts for the indicated times. SSRP1 input is 10% of the total. **c** WB of chromatin isolated from interphase egg extracts supplemented with sperm nuclei pre-incubated with buffer (−) or purified histone H1 (+) at a final concentration of 2 μM for 30 min in the presence of 200 ng/μl recombinant SSRP1 or control buffer and probed with the indicated antibodies. **d** Representative autoradiography of replication time course assay. Sperm nuclei were pre-incubated for 30 min with 200 ng/μl recombinant SSRP1, 2 μM histone H1, or both and then transferred to interphase extracts. Samples were taken at the indicated times after nuclei and α-$^{32}$P-dCTP addition to egg extracts. OD for each lane is indicated. Images in all figures show a typical result of experiments repeated at least three times.

histone H1 we verified their direct interaction. Importantly, we showed that in vitro the SSRP1-SPT16 complex is able to interact with histone H1 and that SSRP1 retains the ability to interact with it even in the absence of SPT16 (Fig. 4a). To identify the molecular determinants required for histone H1 interaction we generated SSRP1 deletion mutants in known SSRP1 domains (Fig. 4b). We found that SSRP1 is able to interact with histone H1 in the presence or in the absence of SPT16 and when the HMG domain was deleted (ΔHMG). SSRP1 protein containing a point mutation (R213D) disrupting a second DNA binding domain[26] was also able to interact with histone H1 (Fig. 4c). The N-terminus domain (NTD) alone, which contains the conserved pleckstrin homology (PH) protein interaction domains PH1 and PH2 involved in hetero-dimerization with SPT16 (ref. [27]), was equally able to bind histone H1 (Fig. 4c). Instead, the N-terminus deletion mutant (ΔNTD), lacking the first 177 aa of the N-terminus completely abolished the binding of SSRP1 to histone H1 (Fig. 4c). These experiments define an important function for the conserved NTD of SSRP1 in binding histone H1 and revealed an additional histone chaperone role for SSRP1, which is already known to interact with histone H2A-H2B and H3-H4 dimers[27]. How these activities are coordinated with histone H1 binding in vivo remains to be investigated.

Next, we tested the ability of the different mutants to promote replication of somatic nuclei, histone H1 eviction and DNA replication origin formation. We found that the ΔNTD mutant protein could not promote somatic nuclei replication (Fig. 5a, b)

and origin assembly (Fig. 5c). Instead, pre-incubation of somatic nuclei with SSRP1 NTD domain was sufficient to stimulate DNA replication (Fig. 5a, b) and origin assembly (Fig. 5c). ΔHMG and R213D mutations, instead, did not affect the ability of SSRP1 to stimulate somatic nuclei replication (Fig. 5a, b) and replication origin assembly (Fig. 5c). These results indicated that SSRP1 NTD was responsible for the stimulation of origin assembly on somatic nuclei. Accordingly, the ΔNTD mutant failed to induce histone H1 eviction from somatic nuclei and to counteract H1-mediated replication origin assembly inhibition (Fig. 5d, e). Instead, the NTD alone could significantly promote the eviction of histone H1 bound to somatic nuclei when pre-incubated with them (Fig. 5f). As the NTD alone is able to bind chromatin (Fig. 5f) it is possible that histone H1 removal is mediated by NTD binding both H1 and DNA, weakening histone H1 interaction with linker DNA.

Although SSRP1 alone is capable of removing histone H1 from chromatin through its NTD we cannot completely rule out the requirement of endogenous SPT16 for this task, as we could not deplete SPT16 with our anti Xenopus SPT16 antibody. This was likely due to the high concentration of SPT16 in egg extract (~620 nM)[20]. However, we believe that the participation of SPT16 in H1 removal is unlikely as SSRP1 is able to bind somatic nuclei in the absence of detectable SPT16 before nuclei incubation in egg extract (Supplementary Fig. 3a). Furthermore, recombinant SSRP1 does not stimulate further recruitment of endogenous SPT16 onto chromatin when somatic nuclei preloaded with SSRP1 are transferred to egg extract (Supplementary Fig. 3b).

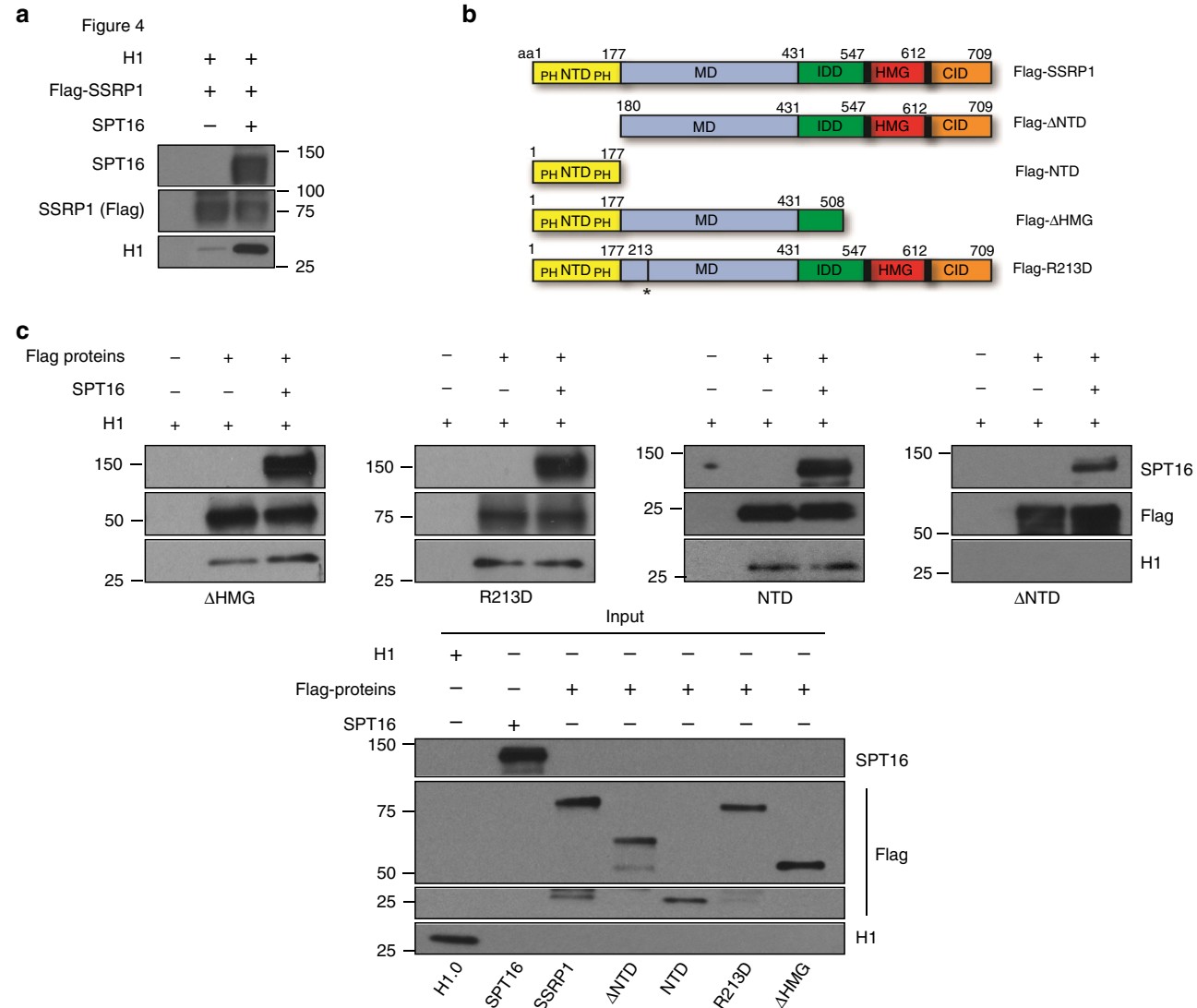

**Fig. 4 SSRP1 interaction with histone H1 through its N terminal domain. a** WB of Flag pull-down performed with recombinant human SPT16, Flag-SSRP1 and purified histone H1 using the indicated antibodies. **b** Schematic representation of Flag-SSRP1 mutated recombinant proteins used. Structural domains of wild-type SSRP1 protein are marked in color: N-terminal domain, NTD (Yellow); intrinsically disordered domain, IDD (green); C-terminal domain, CID (orange); Middle domain, MD (blue); HMG-box (red) and positively charged domains (black). Numbers indicate amino acid residues. PH indicates pleckstrin homology domain. *Indicates mutated residue. **c**) WB of in vitro pull-downs using the anti-Flag antibodies bound to agarose beads incubated with the indicated combinations of recombinant wild-type and mutant proteins. The mutated Flag-SSRP1 proteins used for the pull-downs are indicated under each panel. The inputs are shown in the panel below (Input).

**SSRP1 delays MBT onset**. Having established the roles of SSRP1 and histone H1 in regulating the assembly of replication origins we asked whether these activities affect the physiological regulation of replication origin assembly during embryonic development. Somatic replication cycles are established at MBT, when the number of embryonic replication origin is significantly reduced. This coincides with the elongation of the cell cycle through the downregulation of Cdk1 activity, which is modulated by the inhibitory phosphorylation of tyrosine 15 (pTyr 15)[28]. Phosphorylation of tyrosine 15 reflects the activation of Chk1 at MBT, which contributes to cell cycle progression regulation in post-MBT cycles[28]. Expression of somatic histone H1 proteins H1A and H1B, which are highly similar to histone H1.0, starts around stage 5 and rapidly increases at MBT[29] whereas SSRP1 protein levels, together with SPT16, decrease at this stage (Supplementary Fig. 4) consistent with previous MS analysis[29]. To test whether histone H1 proteins expression and decreasing levels of

SSRP1 protein are determinant for the establishment of somatic replication mode, we injected Myc-tagged SSRP1 mRNA in fertilized eggs and monitored their development following the number of synchronous divisions-only for about 450 min from fertilization until MBT onset, which normally occurs between cycles 12 and 14 (Fig. 6a and Supplementary Movie 1). Strikingly, SSRP1 overexpression induced a sharp delay in the onset of the MBT, as shown by the increase in the number of rapid synchronous divisions from 14 to 15 in all injected embryos compared to buffer injected ones (Fig. 6b). The injection of SSRP1 did not affect the total level of histone H1 protein expressed in post-MBT embryos when compared to buffer injected control embryos at similar stage (Supplementary Fig. 5a). Co-injection of an excess of histone H1, instead, suppressed SSRP1 induced MBT delay (Supplementary Fig. 5b, c).

Consistent with SSRP1-dependent induction of extra synchronous pre-MBT cycles, the DNA content in embryos injected with

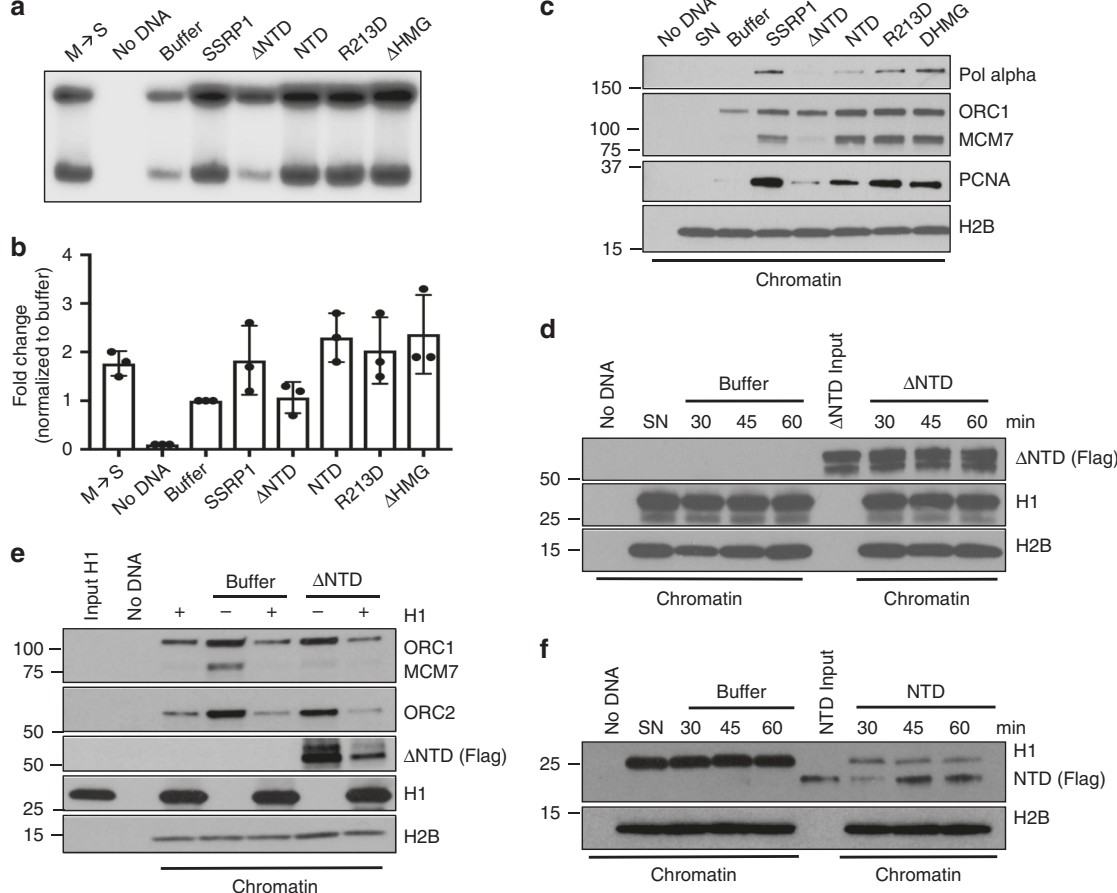

**Fig. 5 SSRP1 N-Terminal domain dependent stimulation of DNA replication through histone H1 chromatin eviction. a** Representative autoradiography of a DNA replication assay showing α-$^{32}$P-dCTP incorporation for 120 min in SN pre-incubated with 200 ng/µl recombinant SSRP1 or equimolar amounts of its mutated versions, as indicated. **b** Graph indicating fold change in somatic DNA replication efficiency relative to buffer considered as 1. SN were pre-incubated with 200 ng/µl recombinant SSRP1 or equimolar amounts of its mutated versions as indicated. Bars represent mean ± SEM. $n = 3$ independent experiments; $p < 0.005$ when comparing all values. Two-way anova. **c** WB of chromatin isolated at 60 min after addition to interphase egg extracts of SN pre-incubated with 200 ng/µl recombinant SSRP1 or equimolar amounts of its mutant versions and probed with the indicated antibodies. **d** WB of chromatin isolated at the indicate times from interphase egg extracts supplemented with SN pre-incubated with Flag-ΔNTD or control buffer. **e** WB of chromatin isolated from interphase egg extracts incubated for 60 min with SN exposed to Flag-NTD or control buffer in the presence or absence of purified histone H1. **f** WB of chromatin isolated at the indicate times from interphase egg extracts incubated with somatic nuclei exposed to Flag-NTD or control buffer.

SSRP1 also increased compared to embryos injected with buffer only (Supplementary Fig. 6). Accordingly, phosphorylation of Cdk1 Tyr15 was delayed (Fig. 6c), especially between stage 8 and 9 in which embryos were sampled every 30 min to better monitor the effect on MBT timing induced by SSRP1. Injection of the mRNA of the ΔNTD mutant, which cannot interact with and evict histone H1 from chromatin, was unable to delay MBT, promote extra synchronous cell divisions (Fig. 6d, e and Supplementary Movie 2), increase the DNA content (Supplementary Fig. 6) or delay Cdk1 Tyr15 phosphorylation (Fig. 6f). Significantly, the NTD alone was instead able to promote extra synchronous cycles and MBT delay (Fig 6g, h and Supplementary Movie 3), increase the DNA content (Supplementary Fig. 6) and delay Cdk1 Tyr15 phosphorylation (Fig. 6i). Of note, the expression of the injected Myc-tagged SSRP1, ΔNTD and NTD proteins persisted in post-MBT embryos suggesting that their action continued during development (Fig. 6c, f, i).

Intriguingly, SSRP1 injection promoted general transcription stimulation of a number of genes normally expressed at MBT as shown by quantitative Polymerase Chain Reaction (qPCR) (Supplementary Fig. 7). SSRP1 dependent transcriptional stimulation started around MBT and continued in post-MBT embryos (Supplementary Fig. 7). Among the stimulated genes there were genes suppressed by histone H1, including xMyoD and BMP4 (ref. [30]) (Supplementary Fig. 7).

**SSRP1 accelerates development by increasing cell cycle speed.** Surprisingly, SSRP1 induced MBT delay did not disturb post-MBT development events. This is contrast with other interventions that stimulate active replication origins formation on somatic nuclei, which lead to embryo death at gastrulation[7]. Strikingly, instead, we observed a marked increase of post-MBT development speed. Faster development in SSRP1-injected embryos was harmonic as all major anatomical structures, including the eye vesicle, the neural tube and the tail were normally formed. The effect of SSRP1 could be easily noticed 48 h post fertilization (pf), when SSRP1-injected embryos appeared to be at stage 42–43, normally reached 72 h pf, instead of stage 32–33, typically reached 48 h pf (Fig. 7a). For a quantitative readout of development speed we monitored tadpole length, showing that at 48 h pf the average length of SSRP1-injected embryos was significantly higher than

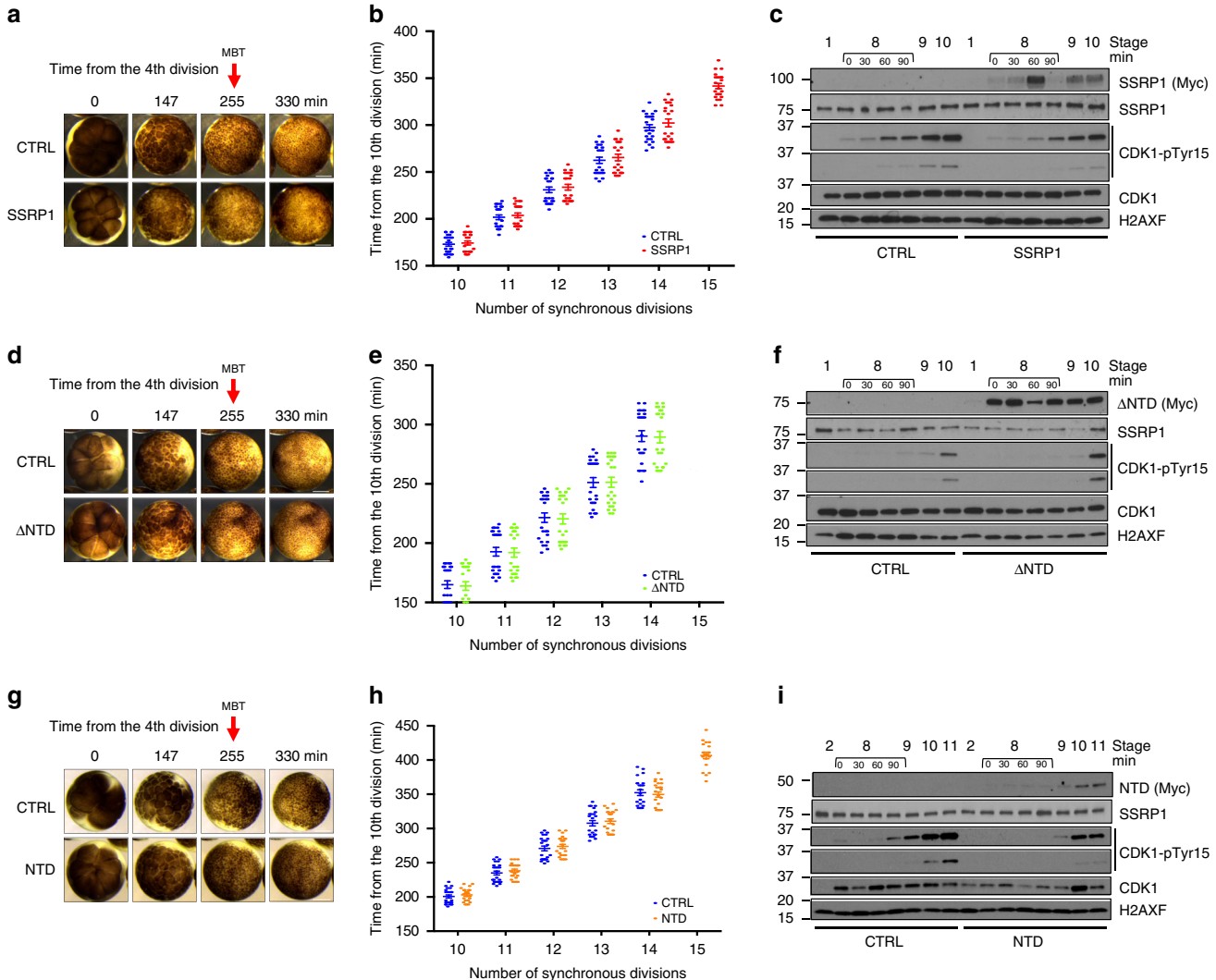

**Fig. 6 SSRP1 induced delay of MBT onset. a** Time lapse video frames taken from movie S1 at the indicated times from the 4th cleavage (set as time zero) of embryos injected at the one-cell stage. Top row: control buffer (CTRL) injected embryos; Bottom row: embryos injected with Myc-SSRP1 mRNA. Size bar = 500 μm. **b** Graph showing the number of synchronous divisions-only of 24 embryos injected with Myc-SSRP1 mRNA or buffer and monitored up to 450 min from fertilization. Only cleavages after the 10th are shown to simplify graph layout. Each dot represents one embryo. Bars show mean ± SEM. **c** WB using the indicated antibodies of whole embryos injected with Myc-SSRP1 mRNA or buffer and taken at the indicated stages. Stage 8 was sampled every 30 min. **d** Images taken as in (**a**) from time-lapse movie S2. Embryos injected with control buffer or Myc-ΔNTD mRNA. Size bar = 500 μm. **e** Graph showing the number of synchronous divisions monitored as in (**b**) of embryos injected with buffer or Myc-ΔNTD mRNA. **f** WB of whole embryos injected with Myc-ΔNTD mRNA or buffer and taken as in (**c**). **g** Images taken as in (**a**) from time-lapse movie S3 of embryos injected with control buffer or NTD mRNA. Size bar = 500 μm. **h** Graph showing the number of synchronous divisions monitored as in (**b**) of embryos injected with control buffer or Myc-NTD mRNA. **i** WB of whole embryos injected with control buffer or Myc-NTD mRNA taken as in (**c**).

control embryos and more typical of tadpoles of more advanced stages (Fig. 7b). SSRP1 induced acceleration persisted throughout post-MBT development and stopped around 96 h pf with embryos readjusting to a normal development speed and size, likely due to the fading of exogenous protein expression.

To understand the mechanism responsible for this accelerated development we measured the duration of post-MBT cell cycles by monitoring different areas of tadpoles with remaining synchronous divisions[31]. In control embryos pre-MBT cycles lasted around 35 min, whereas cycles 13, 14 and 15 progressively extended lasting up to 45–65 min. Cell cycle decelerated consistently at post-MBT cycle 16, the last one that could be monitored by non-invasive imaging, which lasted up to 150 min. Notably, we found that the average length of post-MBT cell cycles

in embryos injected with SSRP1 was strongly reduced, lasting up to 50% less compared to control embryos (Fig. 7c).

In contrast, the expression of ΔNTD mutant protein was unable to promote development acceleration (Fig. 7d, e). Tadpoles exposed to ΔNTD mRNA developed with normal pace and reached similar lengths at 48 h pf with only a slightly negative effect on embryo length. Consistently, the ΔNTD did not affect the duration of the cell cycle (Fig. 7f).

The expression of equivalent levels of mRNA encoding for the NTD was instead able to promote efficient developmental acceleration, which was mostly visible at 24 h pf (Fig. 7g, h) likely due to a higher efficiency towards H1 removal and a more limited stability of the injected NTD fragment. The NTD expression had also marked effects on the average cell cycle

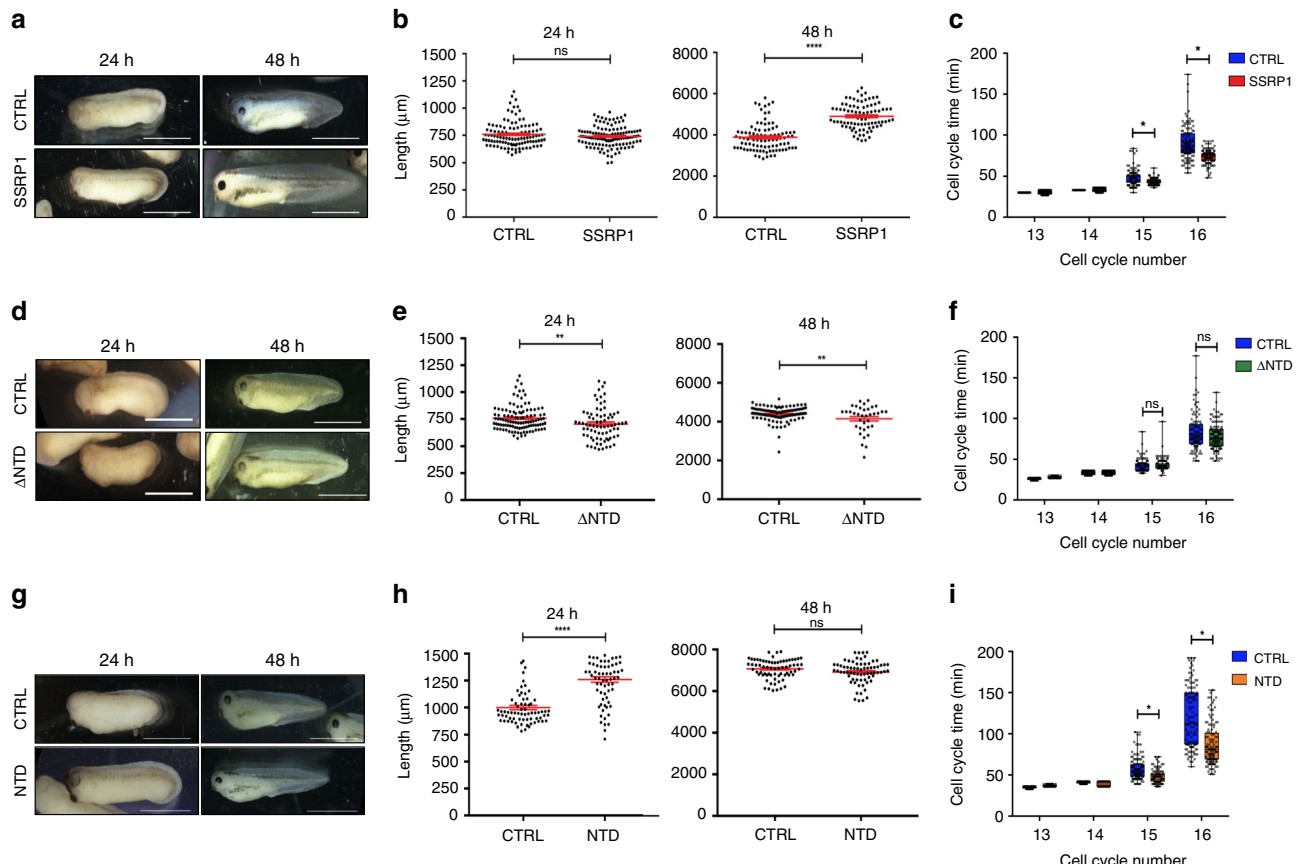

**Fig. 7 SSRP1 induced development acceleration through cell cycle shortening. a** Representative images of control buffer (CTRL) and Myc-SSRP1 (SSRP1) injected *Xenopus laevis* embryos at 24 and 48 hours pf. Size bar = 1000 μm. **b** Graph showing lengths of CTRL and SSRP1- injected embryos measured 24 and 48 h pf. Each point represents one embryo. Bars show mean ± SEM; ****$p < 0.0001$; ns = not significant; $n = 120$ embryos; unpaired two-tailed $t$ test. **c** Graph showing average length of cell cycle post-MBT per embryo measured for 30 cells starting at the 12th cell division using four embryos for each indicated condition. Cell cycle duration times between two sequential divisions were manually annotated from movies; $n = 120$ cell cycles; *$p < 0.01$; unpaired two-tailed $t$-test. **d** Representative images of CTRL and Myc-ΔNTD—injected embryos at 24 and 48 h post-fertilization. Size bar = 1000 μm. **e** Graph showing length of CTRL and Myc-ΔNTD mRNA injected embryos measured 24 and 48 h pf. Each point represents one animal. Bars show mean ± SEM; $n = 100$ embryos; **$p < 0.01$; unpaired two-tailed $t$ test. **f** Graph showing average duration of cell cycle post-MBT measured as in (**c**); $n = 120$ cells; *$p < 0.01$; unpaired two-tailed $t$ test. **g** Representative images of CTRL and NTD- injected embryos 24 and 48 h pf. All images are at identical scale. Size bar = 1000 μm. **h** Graph showing length of CTRL and NTD mRNA injected embryos measured 24 and 48 h pf. Each point represents one embryo. Bars show mean ± SEM; $n = 70$ embryos; ****$p < 0.0001$; unpaired two-tailed $t$ test. **i** Graph showing average duration of cell cycle post-MBT measured as in (**c**); $n = 120$ cell cycles; *$p < 0.01$; unpaired two-tailed $t$ test. For graphs in (**c**), (**f**), (**i**), box height represents the 25th to 75th percentiles and the centerline represents the median. The whiskers extend to the farthest data point.

duration, which was significantly reduced as in the case of SSRP1 injection (Fig. 7i).

**SSRP1 promotes origin assembly and DNA transcription in vivo**. To verify whether the acceleration of the cell cycle in post-MBT embryos by SSRP1 was due to the stimulation of DNA replication we monitored replication dynamics in embryos by DNA combing. To this end we made extracts from post-MBT stage embryos injected with buffer or SSRP1 mRNA (Supplementary Fig. 8a). These extracts contain nuclei that are able to replicate their DNA in vitro in the presence of an ATP regeneration system, reproducing DNA replication dynamics typical of the stage they derived from[32]. We pulse labeled DNA replication for short time using biotin-dUTP and monitored DNA replication initiation events and IODs by DNA combing as in Figure 2d. We observed that replication origins in post-MBT embryos injected with buffer had an average IOD of 17.6 ± 0.7 Kb and a density of ~4–5 origins every 100 Kb (Supplementary Fig. 8b, c), similar to what previously reported[32]. In embryos injected with

SSRP1, instead, we observed a significant stimulation of replication origin assembly with origins placed at an average IOD of 10.1 ± 0.3 Kb and at density of ~8–9 origins every 100 Kb (Supplementary Fig. 8b, c). These results are compatible with reduced length of S-phase and therefore with shorter post-MBT cell cycle duration. Notably, SSRP1 injection was also able to stimulate de novo RNA transcription, which we measured by monitoring ethynyl-uridine (EU) incorporation in RNA synthesized in post-MBT extracts (Supplementary Fig. 8d, e). This stimulation was in agreement with qPCR data of early transcribed genes in SSRP1-injected embryos (Supplementary Fig. 7). As replication origins might interfere with transcription events inducing DNA damage as result of DNA transcription and DNA replication conflicts[33], we monitored the occurrence of DNA double strand breaks by measuring the levels of H2AX phosphorylation[34] in embryos with accelerated development. Although phospho-H2AX levels could be detected in tissue sections of embryos subjected to mild ionizing radiation, SSRP1 did not induce measurable levels of H2AX phosphorylation (Supplementary Fig. 9). These results suggest that stimulation of DNA replication origin assembly by

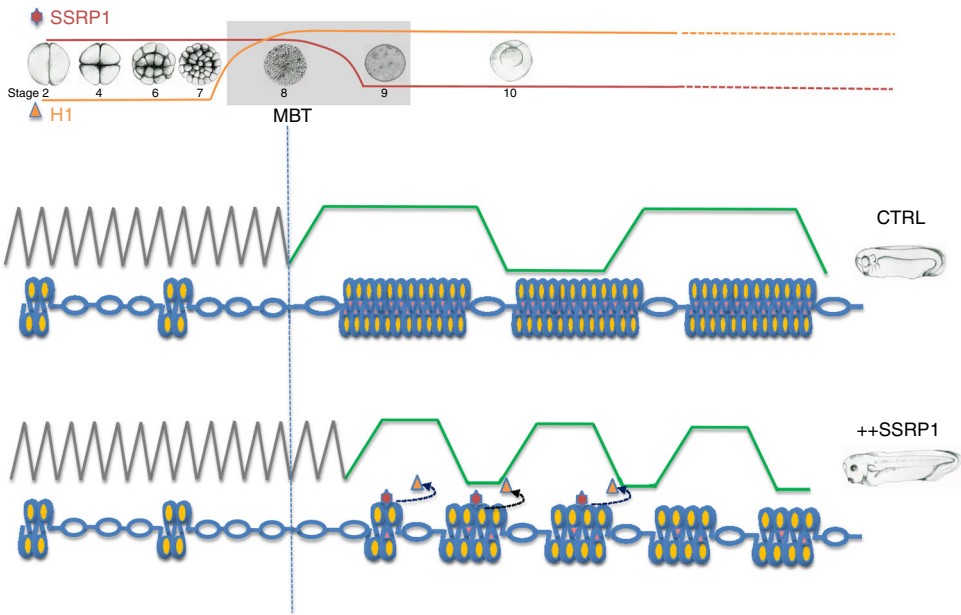

**Fig. 8 Proposed model.** In vitro SSRP1 suppresses histone H1 chromatin association and histone H1-mediated inhibition of DNA replication origins assembly. In vivo SSRP1 levels decrease around MBT when somatic forms of histone H1 start to be expressed. Overexpression of SSRP1 delays MBT onset. Persistent expression of SSRP1 in post-MBT embryos promotes faster somatic cell cycles and accelerated development. See text for more details.

SSRP1 during development does not cause significant DNA damage and is therefore compatible with ongoing transcription, consistent also with the normal morphology of fast developing embryos injected with SSRP1 or NTD.

## Discussion

Our findings indicate that somatic nuclei cannot be efficiently replicated in interphase egg extract, in which replication factors are not limiting, due to a non-permissive status for pre-RC assembly that prevents efficient loading of ORC and MCM complexes onto somatic chromatin. We show that: (1) histone H1 bound to somatic chromatin is responsible for this non-permissive status; (2) SSRP1 promotes the assembly of active replication origins onto somatic chromatin DNA by removing histone H1; (3) the histone H1 binding region of SSRP1 NTD is necessary to counteract histone H1; (4) SSRP1 protein levels decay at MBT; (5) SSRP1 and its NTD domain are able to delay MBT onset; (6) persistent SSRP1 levels promote faster cell divisions and embryonic development acceleration; (7) delayed MBT and accelerated development induced by SSRP1-mediated increase in origin assembly do not lead to DNA damage accumulation.

Non-permissive chromatin status is probably ensured by tight binding of histone H1 to linker DNA forming high order chromatin structures with restricted nucleosome mobility[35] that limit the access of the ORC complex to DNA (Fig. 8). Consistent with this histone H1 in mammalian cells stabilizes nucleosome occupancy on more than 50% of AT rich regions[21], which are preferential sites for ORC complex binding[36]. Therefore, histone H1 removal from somatic chromatin and consequent nucleosome destabilization at these sites by SSRP1 might facilitate ORC complex DNA binding leading to high replication origin assembly typical of embryonic cycles. Somatic histone H1 proteins H1A and H1B, which together with histone H1.0 share common binding mode to linker DNA[37], are expressed around MBT and constitute the major forms of somatic histone H1 at this stage[23]. Embryonic maternal histone H1M, which apparently does not interfere with replication origin assembly, is instead lost at MBT[35]. The presence of high SSRP1 levels before MBT when somatic histone H1 levels start to rise

might counteract premature incorporation of histone H1 into chromatin, avoiding disruption of rapid DNA replication cycles. Parallel SSRP1 decay and histone H1 increase could cooperate to determine the onset of somatic cycles post MBT (Fig. 8). SSRP1 persistence at lower levels might continue to modulate histone H1 incorporation into chromatin throughout development probably at specific loci even after declining. The interaction with histone H1 might take place between the PH domains at the N-terminal region of SSRP1 and the globular portion of histone H1, which binds the DNA at the nucleosome entry-exit point[37]. This could weaken the binding of histone H1 to DNA promoting its release. Alternatively, SSRP1 might interact with the tail of histone H1 disengaging it from the linker DNA[37]. Structural analysis of SSRP1 together with histone H1 is required to discriminate between these models.

Previously, it was shown that the lengthening of the cell cycle and the resulting asynchronous cell division that begins at the MBT is due, at least in part, to a decline in the amounts of four limiting replication factors Cut5, RecQ4, Treslin, and Drf1 (ref. [7]). MBT delay can indeed be obtained by overexpressing these four factors[7]. This, however, leads to embryo death around gastrulation[7]. Surprisingly, persistent high levels of SSRP1 not only promote MBT delay without interfering with normal development but also lead to its dramatic acceleration due to reduced length of DNA synthesis and cell cycle (Fig 8). These observations suggest that replication initiation factors are sufficient to support DNA replication at MBT in the presence of persistent expression of SSRP1.

SSRP1-mediated delay of MBT and accelerated development might be a physiological phenomenon, possibly activated in environmental conditions requiring faster development, a response present in some amphibian species[38]. The faster pace of DNA synthesis imposed by SSRP1 might indirectly impact on the cell cycle delaying the activation of CHK1, which coordinates entry into mitosis with ongoing DNA synthesis at MBT[28] by modulating the levels of Drf1 (ref. [39]) and Cdk1 tyr 15 phosphorylation through Cdc25A[40] (Fig. 8). How faster cell cycle is coupled to rapid cell growth remains to be established.

Unexpectedly, SSRP1-mediated MBT delay and accelerated development due to stimulation of DNA replication does not

interfere with active transcription, as shown by the absence of DNA double strand breaks, which can occur when DNA replication forks collide head on with transcription units[33].

Intriguingly, SSRP1 is enriched at transcription start sites (TSSs)[15] whereas histone H1 is selective depleted from these[41]. Consistent with this arrangement we observe general SSRP1 dependent transcription stimulation at MBT, which might in part be due to histone H1 removal from TSSs. SSRP1-mediated stimulation of replication origin assembly might take place at TSSs, where co-directional activation of replication forks and transcription units could avert head-on collisions, and therefore, DNA damage. Alternatively, SSRP1 itself might help to resolve conflicts between transcription and DNA replication as already shown for the FACT complex in other organisms[42].

Altogether these results reveal a key regulatory epigenetic mechanism that coordinates DNA replication with cell cycle and development in vertebrate organisms. This pathway could be an attractive target for cancer therapy as its modulation might restore normal histone H1 incorporation onto chromatin in SSRP1 overexpressing tumors and in cancer stem cells bearing reduced levels of histone H1 (refs. [16,17]), thus limiting DNA replication and cell proliferation.

## Methods

**Experimental model**. Eggs derived from *Xenopus laevis* frogs were used as experimental model system. Collection of eggs from the female frogs was performed in a non-invasive way following chorionic gonadotropin (Sigma, CG10) injection, complying with all relevant ethical regulations for animal testing and research. Occasional surgical procedures were performed on the male frogs to harvest sperm nuclei. Experimental protocols were approved by IFOM Animal Welfare committee and the Italian Ministry of Health. Part of the experiments also were conducted at Clare Hall Laboratories, London Research Institute (LRI) under LRI and UK Home Office authorization for the use of amphibians. The number of animals used was kept to a minimum and was calculated taking into account the number eggs required to obtain the cytoplasmic extract needed for the experiments described. The animals were kept in highly regulated and monitored conditions with room and water temperature at 19 °C. Basic husbandry requirements were provided by the IFOM *Xenopus* facility and LRI amphibians facility.

**Preparation of egg and embryo extracts**. *Xenopus* mitotic egg extracts (CSF extracts) were prepared as previously described[43]. Briefly, eggs were laid and collected in MMR solution (100 mM NaCl, 2 mM KCl, 1 mM MgCl$_2$, 2 mM CaCl$_2$, 0.1 mM EDTA, 5 mM HEPES, pH 8). The jelly coat of eggs was removed by incubation in 2% cysteine in Murray salt solution (100 mM KCl, 1 mM MgCl$_2$, 0, 1 mM CaCl$_2$). The dejellied eggs were then washed three times with XB wash buffer (100 mM HEPES-KOH, pH 7.8, 500 mM sucrose) and three times with CSF-XB buffer (100 mM KCl, 5 mM EGTA, 2 mM MgCl$_2$). Wash buffer was then poured off and 10 mg/ml of cytochalasin B (SIGMA) in DMSO and LPC protease inhibitors (30 mg/ml each of leupeptin, pepstatin and chymostatin) were added. Eggs were packed for 1 min at 1000 × *g* in a swing bucket rotor at 16 °C. Excess liquid on top of the packed eggs was removed and the eggs were crushed by centrifugation in a swing rotor at 20,000 × *g* for 14 min. The resulting cytoplasmic extract (middle golden yellow layer) was removed by puncturing the side of the tube with a 19-gauge needle and slowly removing the cytoplasmic layer with a 2 ml syringe. This extract was placed in a 5 ml polypropylene round-bottomed tube (Falcon 2063). Cycloheximide 0.2 mg/ml (Calbiochem) was then added. Energy mix (375 mM creatine phosphate, 50 mM ATP, 10 mM EGTA, 50 mM MgCl$_2$) was added at 1:50 dilution; LPC protease inhibitors mix (30 mg/ml each of leupeptin, pepstatin and chymostatin) and cytochalasin B (10 mg/ml) were added at 1:1000 dilution. The extract was then gently mixed and centrifuged for 15 min at 4 °C. The resulting extract was removed by syringe as above, and placed in a new tube ready for use. CSF egg extracts were induced to enter interphase by addition of 0.4 mM CaCl$_2$ and used as indicated in text and figure legends.

Embryonic nucleo-cytoplasmic extracts were prepared as described[32] with some modifications. Briefly, 400 dejellied late blastula (stage 9) embryos were washed in embryo extraction buffer (10 mM Hepes, pH 7.7 KOH, 100 mM KCl, 50 mM Sucrose, 1 mM MgCl$_2$, 0.1 mM CaCl$_2$ and spun at 700 × *g* for 1 min to pack embryos without crushing. Excess liquid was removed and embryos were centrifuged at 17,000 × *g* for 10 min at 4 °C. The two middle phases (yellow containing cell cytoplasm and gray containing the nuclei) were isolated with a needle and combined.

**DNA templates**. Demembranated sperm nuclei were prepared as previously described[44]. Briefly, *X. laevis* males were primed with 50 U PMSG (Folligon) 7 days

in advance and with 300 U HCG the day before testis dissection. Testis were isolated and collected in EB buffer (50 mM HEPES-KOH, pH 7.6, 50 mM KCl, 5 mM MgCl$_2$, 5 mM EGTA, 2 mM β-mercaptoethanol), rinsed three times in ice-cold EB buffer and finely chopped with a razor blade. The material was then transferred to 15 mL Falcon tube and spun for 5 min in a swinging-bucket rotor at 4250 × *g* at 4 °C. The pellet was then resuspended in 1.5 mL of SuNaSp buffer at room temperature (0.25 M sucrose, 75 mM NaCl, 0.5 mM spermidine, 0.15 mM spermine). To remove membranes lysolecithin 2 mg/mL (SIGMA) was added and incubated for 10 min at room temperature. Reaction was stopped by adding 3% BSA (SIGMA). The pellet was resuspended again in 1 mL ice-cold EB and spun at 2000 × *g* for 5 min at 4 °C. The final pellet was resuspended in 400 µl of EB suplemebted with 30% glycerol.

Erythrocyte somatic nuclei were purified from two female frogs. Blood was collected in 10 ml of Barth solution (88 mM NaCl, 2 mM KCl, 0.82 mM MgCl$_2$, 10 mM Hepes pH 7.4) and, immediately after, diluted into 10 ml ice-cold 0.6×SSC buffer (150 mM NaCl, 15 mM Na citrate, pH 7.0) supplemented with heparin (10 mg/ml). The diluted blood was underlaid with 0.5 volumes of Ficoll 5.7% w/v in 0.6 × SSC-heparin solution and centrifuged at low speed for 15 min at 4 °C. The red pellet was resuspended in 0.6 × SSC and centrifuged through Ficoll. The resulting pellet was washed three times in nuclear isolation buffer (NIB) (250 mM sucrose, 25 mM NaCl, 10 mM PIPES, 1.5 mM MgCl$_2$, 0.5 mM spermidine, 0.15 mM spermine) at pH 7.0 and resuspended at 1 × 10$^8$ cells/ml. For demembranation, erythrocyte somatic nuclei were diluted to 4 × 10$^7$ cells/ml in NIB at 23 °C. An equal volume of NIB containing 40 µg/ml lysolecithin and two volumes of trypsin type XIII (1 µg/ml) solution (SIGMA) were added. After 5 min, 10 µg/ml soy bean trypsin inhibitor (SIGMA) and 0.4% BSA were added. Nuclei were centrifuged at 1000 × *g*, washed twice with NIB and resuspended in NIB at 200,000 nuclei/µl, supplemented with 30% glycerol and frozen in small aliquots in liquid nitrogen.

**Chromatin isolation from egg extract**. To isolate chromatin fractions, sperm or erythrocyte nuclei (4000 nuclei/µl) were added to 30 µl egg extracts and incubated at 23 °C for the indicated time. Samples were diluted with 10 volumes of ice-cold EB (100 mM KCl, 2.5 mM MgCl$_2$ and 50 mM HEPES-KOH pH 7.5) containing 0.25% NP-40 and centrifuged through a 0.5 M sucrose layer at 10,000 × *g* for 5 min at 4 °C. Pellets were washed once with EB and resuspended in SDS-PAGE sample buffer. Proteins were then resolved by SDS-PAGE and monitored by WB.

**Replication reactions**. DNA replication in *Xenopus* egg extract was performed as previously described[34]. Briefly, sperm or erythrocyte nuclei (400-8000 nuclei/µl) were added to 20 µl interphase egg extract supplemented with 0.1 µl of α-$^{32}$P-dCTP (250 mCi; 3000 Ci/mmol) and incubated at 23 °C. Reactions were stopped with Stop buffer (8 mM EDTA, 80 mM Tris pH 8.0, 1% w/v SDS), supplemented with 1 mg/ml Proteinase K and incubated at 50 °C for 2 h. Samples were treated with RNase (0.6 mg/ml) to degrade any trace of RNA. Genomic DNA was phenol/ethanol exracted and then separated from unincorporated nucleotides by electrophoresis through a 0.8% agarose gel. The gel was fixed in 30% TCA for 20 min, dried and exposed for autoradiography. For quantification of DNA replication, the gels were exposed to a phosphoscreen (GE Healthcare). The recorded radioactive signal was monitored within a phosphoImager (Typhoon) and measured with ImageQuant software. Alternatively, absolute amount of replicated DNA was quantified as previously described[45–47]. Briefly samples were precipitated with 4 ml of 4 °C 10% TCA for at least 30 min at 4 °C. A fraction of the precipitated sample (40 µl) was spotted spotted on paper filter for measurment of total $^{32}$P. The remainder of the sample was filtered through a glass fiber filter, which was washed twice with 8 ml of 4 °C 5% TCA and then with 8 ml of ethanol. Each filter emission was quantified through a beta-counter. Quantification of replicated DNA was obtained by dividing the $^{32}$P incorporated into DNA captured on the glass fiber filter by the total $^{32}$P on the paper filter, yelding the percentage of total $^{32}$P incorporated into DNA. For embryo nuclei replication monitoring through DNA combing (see below) nucleo-cytoplasmic lysates were supplemented with energy mix (7.5 mM creatine phosphate, Creatine kinase 1 mM ATP) and 20 µM biotin-dUTP to monitor DNA replication at 23 °C for 6 min.

**RNA transcription labeling in *Xenopus* embryos**. To label de novo synthesis of RNA in post-MBT Xenopus embryos embryonic nucleo-cytoplasmic extracts were prepared as described above. Click-iT RNA Alexa Fluor 594 (C10330) and Alexa Fluor 488 (C10329) imaging kits were adapted for application in *Xenopus* embryonic lysates. Briefly, click-iT EU (5-ethynyl Uridine) was added to *Xenopus* embryonic lysates at a final concentration of 1 mM for 30 min. Total RNA was extracted by applying TRIzol reagent to lysates (Invitrogen) followed by Direct-zol™ DNA/RNA MiniPrep kit (Zymo Research) according to manufacturer instructions. Then, 10 µg EU-labeled RNA were incubated with a 1×working solution of click-iT reaction cocktail, containing the Alexa Fluor 594 azide and CuSO$_4$ for 1 h in the dark at room temperature. The reaction was repeated with the Alexa Fluor 488 azide to evaluate consistency of labeling. The labeled RNA was precipitated with ethanol and re-suspended in nuclease-free water. Samples were separated on 1% TAE agarose gel and total RNA was visualized with ethidium bromide staining to monitor RNA quality. Total fluorescence of labeled RNA in each lane was detected and quantified with fluorescence detection system (Bio-rad).

**Extract fractionation and protein identification**. Egg extract was fractionated by differential precipitation with PEG as previously described with some modifications[48]. Briefly, extract was diluted fourfold in LFB buffer (40 mM Hepes-KOH pH 8.0, 20 mM K₂HPO₄-KH₂PO₄ pH 8.0, 2 mM MgCl₂, 1 mM EGTA, 2 mM DTT, 10% w/v Sucrose, 1 μg/μl each of Leupeptin, Pepstatin, Aprotin). The diluted extract was centrifuged for 40 min at 80,000 × g in a swing out rotor at 4 °C. The supernatant was carefully transferred to avoid pellet contamination and supplemented with 50% PEG 6000 solution (SIGMA) to give a final concentration of 9% PEG. Samples were incubated on ice for 30 min and precipitated proteins removed by spinning at 10,000 × g for 10 min in a fixed-angle rotor. The supernatant was recovered and the pellet resuspended in LFB at 5× concentration with respect to undiluted extract. After collecting the aliquote, the supernatant was used for subsequent precipitations. Active fractions were further separated on phospho-celluose column (Whatmann P11) equilibrated with 0.1 M NaCl, 20 m HEPES-KOH (pH 7.9), 0.5 m EDTA, 10 mM DTT, 10% glycerol and eluted with NaCl gradient in the same buffer. Each fraction was then tested for its ability to induce replication of somatic erythrocytes nuclei in interphase extract. Each replication reaction was performed by pre-incubating 4 μl of each fraction with erytrocyte nuclei for 30 min at 23 °C. Reaction was then supplemented with 15 μl of interphase extract and incubated for 120 min in the presence of α-³²P-dCTP. Active fractions were subjected to protein identification by mass spec analysis.

**Mass spectrometry**. Approximately 50 μg of proteins from active fractions were transferred to YM-10 Microcon filters (Cat No. MRCF0R010, Millipore) and centrifuged at 14,000 × g for 20 min. 60 μl of 8 M urea in 0.1 M Tris-HCl, pH 8.5 (UT) were added and the sample was reduced by adding 10 μl of 100 mM DTT for 30 min at room temperature. Samples were centrifuged at 14,000 × g for 20 min. Then, 60 μL UT and 10 μL of 55 mM iodoacetamide were added to the filters and incubated in the dark for 20 min. Filters were washed twice with 100 μL of UT. 1.2 μg of Lys-C (Wako) were added and incubated overnight at room temperature. Subsequently, 100 Tris-HCl, pH 8.5 were added to dilute urea concentration and 0.6 μg of trypsin were added in 100 μL 0.1 M Tris-HCl, pH 8.5. Samples were incubated at room temperature for 6 hours, centrifuged at 14,000 × g for 20 min and finally 50 μL of 0.5 M NaCl were added to the filter and the released peptides were collected by centrifugation. The resulting peptides were desalted on a C18 home made StageTip. Peptides were resuspended in 60 μL of Buffer A (2 % ACN, 0.1% formic acid) and 1 μL was injected for every technical replica. Peptides separation was achieved with a linear gradient from 95% solvent A (2% ACN, 0.1% formic acid) to 40% solvent B (80% acetonitrile, 0.1% formic acid) over 30 min and from 40% to 100% solvent B in 2 min at a constant flow rate of 0.25 μL/min, with a single run time of 35 min.

MS data were acquired using a data-dependent top 12 method, the survey full scan MS spectra (300–1750 Th) were acquired in the Orbitrap with 70000 resolution, AGC target 1e6, IT 120 ms. For HCD spectra resolution was set to 35000, AGC target 1e5, IT 120 ms; normalized collision energy 25% and isolation width of 3.0 m/z. MS data were acquired using a data-dependent top10 method for HCD fragmentation. Survey full scan MS spectra (300–1750 Th) were acquired in the Orbitrap with 70000 resolution, AGC target 1e6, IT 120 ms. For HCD spectra resolution was set to 35000, AGC target 1e5, IT 120 ms; normalized collision energy 25% and isolation width 3.0 m z-1. Raw data were processed with MaxQuant version 1.5.1.2. For protein identification the raw data were processed using Proteome Discoverer (version 1.4.0.288, Thermo Fischer Scientific). MS2 spectra were searched with Mascot engine against custom *Xenopus laevis* revised version (according to Tikira Temu, Matthias Mann, Markus Räschle, Jürgen Cox, Homology-driven assembly of NOn-redundant protEin sequence sets (NOmESS) for MS, with the following parameters: enzyme Trypsin, maximum missed cleavage 2, fixed modification carbamidomethylation (C), variable modification oxidation (M) and protein N-terminal acetylation, peptide tolerance 10 ppm, MS/MS tolerance 20 mmu. Peptide Spectral Matches (PSM) were filtered using percolator based on q-values at a 0.01 FDR (high confidence). Scaffold (version Scaffold_4.3.4, Proteome Software Inc., Portland, OR) was used to validate MS/MS based peptide and protein identifications. Peptide identifications were accepted if they could be established at >95.0% probability by the Peptide Prophet algorithm with Scaffold delta-mass correction. Protein identifications were accepted if they could be established at >99.0% probability. Protein probabilities were assigned by the Protein Prophet algorithm. Proteins that contained similar peptides and could not be differentiated based on MS/MS analysis alone were grouped to satisfy the principles of parsimony. Proteins sharing significant peptide evidence were grouped into clusters. MS was initially performed at Clare Hall Laboratories and then validated in IFOM.

**DNA combing**. For somatic nuclei (4000 n/μl) DNA replication reactions were carried out in egg extract supplemented with 1/20 of energy regeneration mix and 1/50 of cycloheximide solution described above and 20 μM digoxigenin-11-dUTP (Roche) for 75 min in the presence of 5 μg/μl aphidicolin. For sperm nuclei (8000 n/μl) DNA replication reactions were carried out for 10 min in the presence of 20 μM biotin-16-dUTP (Roche). For embryonic nuclei embryonic lysates containing nuclei isolated from 200 dejellied blastula embryos (stage 9) injected at 1-cell stage with buffer or Myc-SSRP1 mRNA, as described above, were incubated for 6 min in the presence of 20 μM biotin-16-dUTP. DNA replication reactions

were stopped by adding 10 volumes of ice-cold 1× PBS buffer. Nuclei were then pelleted on a sucrose cushion by centrifugation at 1000 × g at 4 °C for 7 min. Nuclei pellets were resuspended in 50 μl of PBS, mixed with 50 μl 1% low-melting-point agarose (Lonza) and transferred to a casting mould to prepare the plugs. Plugs were then treated with 2 mg/ml of proteinase K (Roche) in digestion buffer (0.5 M EDTA, pH 8, 10% sarkosyl) at 50 °C overnight. Treatment was repeated the day after with freshly prepared proteinase K solution. Subsequently, plugs were washed four times for 1 h each wash in TE buffer supplemented with 50 mM EDTA. The TE buffer was replaced with 50 mM MES (pH 5.7) (Sigma-Aldrich, M5287 and M5057) and plugs were incubated at 65 °C for 15 min. Once melted, the plugs were treated with 4 units of β-agarase (New England Biolabs) at 42 °C overnight. DNA from the resulting solution was combed on silanized slides (Genomic Vision) at a constant speed of 18 mm/min using the combing apparatus (Genomic Vision). Slides were then dried at 65 °C for 30 min. DNA was fixed onto the slides by a 5 min incubation in methanol:acetic Acid (3:1) and denatured in 2.5 M HCl for 30 min. Slides were sequentially incubated in 70%, 90% and 100% EtOH for 3 min each time, washed three times for 5 min in PBS, dried and blocked for 1 h in BlockAid™ blocking solution (B10710, Invitrogen) at 37 °C. Digoxigenin was detected with mouse anti-digoxigenin FITC antibody (Sigma-Aldrich) used at 1:20 dilution for 1 h at 37 °C. Biotin was detected with Streptavidin Alexa Fluor 594-conjugated antibody (S-11227, Thermo Fisher Scientific) used at 1:20 dilution for 1 h at 37 °C, followed by anti-streptavidin biotinylated antibody (BA-0500, Vector Laboratories) used at 1:20 dilution for 1 hr at 37 °C. This procedure was repeated twice, then followed by staining with an anti-ssDNA antibody (MAB3034, MerckMillipore) used at 1:300 dilution for 1 h at 37 °C. Secondary antibodies Alexa Fluor 488 conjugated rabbit anti-mouse (A-11059, Thermo Fisher Scientific) followed by Alexa Fluor FITC conjugated goat anti-rabbit for signal enhancement (F-2765, Thermo Fisher Scientific) were used at 1:50 with each incubation lasting 1 h. In between all antibody incubations, slides were washed twice for 2 min in PBS + 0.05% Tween 20. All antibodies were diluted in BlockAid™ blocking solution. Finally, the slides were washed three times with PBS, 0.05% Tween 20, once again in PBS, and then mounted on a microscope slide with Mowiol mounting medium (Sigma-Aldrich). Image acquisition was performed with a fully motorized microscope equipped with a camera and controlled by MetaMorph (Universal Imaging Corporation). Inter-origin distances between Dig-dUTP tracks or biotin-dUTP tracks, were measured using the MetaMorph software, and micrometer values were expressed in kilobases using as conversion factor 1 μm = 2.59 kb.

**Plasmid construction**. Human SSRP1 cDNA corresponding to clone NM_003146 was cloned into the pcDNA5 FRT/TO vector (Invitrogen™), modified with N-terminal Flag-TAG (gift from Zuzana Horejesi, Clare Hall Laboratories). DNA encoding full length Flag-SSRP1 1-709aa, Flag-SSRP1^NTD 1-180aa, Flag-SSRP1^ΔHMG 1-508aa, and Flag-SSRP1^ΔNTD 529-709aa were amplified by PCR using primers listed below. The Flag-SSRP1^R213D and Flag-SSRP1^ΔHMG were generated by site-directed mutagenesis with the QuikChange strategy (Stratagene) according to the manufacturer's protocol. The DNA fragments obtained were inserted into the pGEX-4T-3 vector (GE, 28-9545-52). This vector encodes an Glutathion-S-Transferase (GST)-tag followed by a Thrombin protease cleavage site.

Primers to clone hSSRP1 cDNA into pcDNA5FRT/TO modified with N-terminal Flag-TAG

hSSRP1-attB1-Fwd
5′GGGGACAAGTTTGTACAAAAAAGCAGGCTTCATGGCAGAGACACTGGAGTTCAACGACG-3′

hSSRP1-attB2-Rev
5′GGGGACCACTTTGTACAAGAAAGCTGGGTCCTACTACTCATCGGATCCTGACGCTGAGTCC3′

Primers to clone hSSRP1 and its mutated versions into pGEX-4T-3 vector

hSSRP1-Full length-Fwd
5′-TTTTCCCGGGACCGGTTTATGGACTACAAGGACGACGATG- 3′

hPmeI-Full length-Rev
5′-TTTTGCGGCCGCGTTTAAACACCACTTTGTACAAGAAAGCTGGG- 3′

NTD-Rev
5′-TTTTGCGGCCGCGTTTAAACGGCCTCAACAGGGTCCACAC- 3′

ΔNTD-Fwd
5′-TTTTGACTCCATGGTTTGCCCAGAATGTGTTGTCAAAGGC- 3′

**Preparation of recombinant proteins**. Glutathion-S-Transferase (GST)-Flag-tagged human SSRP1 protein was expressed in Rosetta™(DE3)pLysS Competent Cells (Novagen, 70956) and purified on Glutathione Sepharose™ 4 Fast Flow (GE, 17513201). Briefly, *E. coli* were grown in LB media containing ampicillin at 37 °C until reaching an OD of 0.6–1.0 at 600 nm. Then cultures were shifted to 25 °C and induced with 0.2 mM isopropyl-β-D-thiogalactopyranoside (IPTG) for 3 h. Cells were collected by centrifugation at 4000 × g for 20 min at 4 °C and pellets were suspended in lysis buffer (300 mM NaCl, 200 mM HEPES pH 7.5, 1% Triton-X100, 1 mM DTT) supplemented with protease inhibitor cocktail (Millipore). Cells were then disrupted by sonication and incubated 10 min at 4 °C with 25 μg/ml DNase I. Lysate was clarified by centrifugation at 21,000 × g for 45 min at 4 °C and incubated with 2 ml of Glutathione beads (GE) in rotation for 1 h at 4 °C. Beads were applied on a gravity column and washed twice with 20 volumes of lysis buffer. The proteins were eluted with elution buffer (150 mM NaCl, 20 mM HEPES pH 7.5, 1 mM DTT

and 20 mM Glutathione pH 8.0) and dyalized. SSRP1 proteins were cleaved from the GST fusion using the appropriate protease, and further purified by SEC on Superdex-200 column (GE Healthcare) pre-equilibrated in SEC buffer (50 mM HEPES, 150 mM NaCl, 10% glycerol). Relevant fractions were concentrated in 50 kDa molecular mass cut-off Amicon ultra centrifugal filters (Millipore). Human Spt16 protein was a kind gift from P. Cherepanov (Imperial College, London, England).

**Immunoprecipitation assays**. Mouse monoclonal anti-SSRP1 antibodies (clone 10D7; abcam) (4 μg) were incubated with 35 μl slurry of Protein A Sepharose® 4 FastFlow (GE Healthcare) for 2 h on a rotating wheel at 4 °C with 1 μg recombinant histone H1.0 (New England Biolabs, M2501S), 5 μg Flag-SSRP1 or Spt16 and Flag-SSRP1 (5 μg each). Beads were washed four times with washing buffer (10 mM Tris-HCl pH 7.4, 150 mM NaCl, 1 mM EDTA, 1 mM EGTA pH 8.0, 1% Triton X-100, 0.2 mM sodium orthovanadate) supplemented with protease inhibitors cocktail. Samples were eluted by heating Laemmli buffer without β-mercaptoethanol for 10 min at 37 °C. Eluted samples were boiled for 5 min, loaded on a SDS-PAGE gel and transfer to nitrocellulose membrane for western blot analysis.

For the in vitro interaction, 1 μg of recombinant histone H1.0 was either incubated with 5 μg of Flag-SSRP1, Flag-SSRP1 mutants or Spt16 and Flag-SSRP1 (5 μg each) in the FLAG-IP buffer (50 mM Tris-HCl pH 7.4, 200 mM NaCl, 1 mM EDTA, 1% Triton X-100) supplemented with fresh protease inhibitors (Millipore) for 2 h on a rotating wheel at 4 °C. FLAG-IP was carried out using anti-FLAG M2 affinity gel (Sigma, A2220). For a single reaction, 40 μl gel suspension was washed twice with FLAG-IP buffer and was incubated with recombinant proteins for 1 h at 4 °C. Protein-bound beads were then washed four times in the same buffer and proteins were eluted by boiling in Laemmli buffer and subjected to immunoblot analysis.

**Xenopus laevis embryos and microinjections**. X. laevis embryos were obtained by in vitro fertilization of freshly laid X. laevis eggs with crushed X. laevis testes. Only batches with greater than 90% fertilization efficiency were used. 20 min after fertilization, embryos were de-jellied in 2% Cysteine pH 8.1 dissolved in 0.1 × MBS (88 mM NaCl, 1 mM KCl, 0.4 CaCl$_2$, 0.33 mM Ca[NO$_3$]$_2$, 0.8 mM MgSO$_4$, 5 mM TRIS-HCl, 2.4 mM NaHCO$_3$). Microinjections were performed using calibrated needles and embryos equilibrated in 1xMBS/3% Ficoll PM-400 (Sigma). Micro-injection needles were generated from borosilicate glass capillaries (Harvard Apparatus, GC 100F-15) using the micropipette puller Sutter p97. Approximately 4 nl mRNA were injected into de-jellied embryos at the 1-cell stage using the microinjector PicoSpritzer III (Parker). After the first cleavage, the buffer was replaced with 1 × MBS/2% Ficoll then 0.1×MBS and embryos were allowed to develop to the desired stage. Embryos were staged according to Nieuwkoop and Faber, 1975. Histone H1.0 cDNA corresponding to human transcript NM_0005318.3 cloned into pET28a(+) vector was obtained from GenScript (OHu16206c). For in vitro transcription, the fragments were cloned between the FseI and AscI sites of pCS2 vector (pCS2-6xMYC_SSRP1, pCS2-6xMYC_SSRP1ΔNTD, pCS2-6xMYC_SSRP1NTD, pCS2-6xMYC_H1.0). To obtain sense RNA from these constructs, plasmids were linearized with NotI; pCS2-6xMYC_H1.0 plasmid was linearized with MfeI. Following linearization, mRNA was expressed from the SP6 promoter using the mMessage mMachine kit (Ambion). All constructs were verified by DNA sequencing. The pCS2 vector was a gift from Philip Zegerman (The Gurdon Institute, University of Cambridge).

Primers for cloning into pCS2 vector:
SSRP1_FseI_Fwd 5′-ATTTTAAAGGCCGGCCAATGGCAGAGACACTG GA- 3′
ΔNTD_FseI_Fwd 5′-ATTTTAAAGGCCGGCCAATGGCATTTGCCCAGA ATGTG- 3′
SSRP1_AscI_Rev 5′-TAAAAGGGGGCGCGCCCTACTCATCGGATCCTG- 3′
NTD_AscI_Rev 5′-TAAAAGGGGGCGCGCGTTTAAACGGCCTCAACAGG GTCC - 3′
H1.0_FseI_Fwd 5′-ATTTTGGGGGCCGGCCAATGACCGAGAATTCCA CGTCCG- 3′
H1.0_AscI_Fwd 5′-TAAAAGGGGGCGCGCCTTACTTCTTCTTGCCGG CCC- 3′

**Xenopus embryos protein analysis**. Embryos were lysed in 50 μl/embryo lysis buffer (50 mM Tris pH 7.6, 150 mM NaCl,10 mM EDTA, 1% TritonX-100) supplemented with protease inhibitors, incubated on ice for 30 min, sonicated for 15 min on High setting 30 s ON/OFF and centrifuged at high speed for 10 min at 4 °C. Protein concentration was determined using Biorad protein assay according to the manufacturer's instructions (Biorad Laboratories).

**Antibodies and western blot analysis**. Samples were resolved by 4–15% SDS-PAGE and analyzed by standard WB techniques. Blots were probed using the following custom antibodies at dilutions indicated in parenthesis: rabbit polyclonal anti-Xenopus Cdc45 (1:2000), mouse monoclonal anti-Xenopus Orc1 (1:100,000) previously described[45,46]; rabbit polyclonal anti-Xenopus Orc2 (1:4000), mouse

monoclonal anti-Cdk1 (A17, 1:500), mouse monoclonal anti-Cdk1 pTyr15 (1:1000), mouse monoclonal anti-Xenopus Cyclin B2 (X121, 1:500) antibodies obtained from J. Gannon, The Francis Crick Institute, London, UK and rabbit polyclonal anti-Xenopus H2A.X-F (1:10,000), obtained by D. Shechter, A. Einstein Institute, New York, NY, USA. The following commercial antibodies were also used: mouse monoclonal anti-MCM7 (47DC141, Santa Cruz Biotechnology, sc-9966, 1:4000), anti-PCNA (PC10, Serotec, MCA1558, 1:5000), mouse monoclonal anti-Cdt1 (F-6, Santa Cruz Biotechnology, sc-365305, 1:500), mouse monoclonal anti-Ssrp1 (10D7, Abcam, ab26212, 1:1000), rabbit polyclonal anti-Ssrp1 (A303-068A, Bethyl Laboratories, 1:1000), rabbit polyclonal anti-Spt16 (H-300, Santa Cruz Biotechnology, sc-28734, 1:250), rabbit polyclonal anti-Cdc6 (Santa Cruz Biotechnology, sc-8341, 1:1000), mouse monoclonal anti-Xenopus Polα (p180) (Abmart, clone 13026-1-3/C199, 1:500), mouse monoclonal anti-H1.0 (Abcam, ab11079, 1:1000), rabbit polyclonal anti-H2B (Millipore, 07-371, 1:1000), mouse monoclonal anti-FLAG-M2-Peroxidase-HRP (A8592, Sigma, 1:1000) and rabbit polyclonal anti-SMC2 (Bethyl, A300-056A, 1:1000). Detection with secondary antibodies was commonly carried out at 1:10,000. Proteins were detected by ECL detection reagents (GE) or WesternBright ECL (Advansta) on Amersham Hyperfilm (GE) or Kodak.

**Immunohistochemistry on Xenopus embryos**. Embryos at the indicated developmental stage were fixed in 4% paraformaldehyde (PFA) overnight at 4 °C. Where indicated tadpoles were irradiated with 5 Gys with Faxitron before fixation. After fixation, embryos were embedded in 1% LMP agarose and then in paraffin with Diapath automatic processor (Diapath). To assess histological features hematoxylin/eosin staining (Diapath) was performed according to standard protocol and samples were mounted in Eukitt (Bio-Optica). For IHC analysis, paraffin was removed with xylene and the sections were re-hydrated in graded alcohol. Antigen retrieval was carried out using pre-heated target retrieval solution for 45 min. Tissue sections were stained with 0.3% H$_2$O$_2$ for 10 min at room temperature for quenching of endogenous peroxide activity, and then blocked with 0.2% Fetal Bovine Serum (FBS) in PBS supplemented with 1% BSA for 1 h and incubated for 2 h with primary antibodies. Mouse monoclonal anti-phospho-histone H2A.X (Ser139) (JBW301, Millipore, 1:500) was used for staining. Antibody binding was detected using a polymer detection kit (GAR-HRP and GAM-HRP, Microtech), followed by a diaminobenzidine chromogen reaction (Peroxidase substrate kit, DAB, SK-4100; Vector Lab). All sections were counterstained with Mayer's hematoxylin and visualized using a bright-field microscope.

**DNA quantification in Xenopus embryos**. For each condition, four embryos were homogenized in 500 μl DNA extraction buffer (10 mM Tris pH 8.0, 0.2 mM EDTA, 0.5% SDS) supplemented with 100 μg/ml proteinase K and incubated at 55 °C overnight. Samples were extracted twice in one volume of phenol:chloroform:iso-amyl alcohol (25:24:1) and once with chloroform. Genomic DNA was ethanol precipitated and resuspended in 15 μl H$_2$O. Samples were incubated with 50 μg/ml RNase A and incubated at 37 °C for 2 h. Genomic DNA was run on 1% TAE agarose gel and visualized by staining with ethidium bromide. DNA was quantified using Quantity One software (BioRad) or ImageJ.

**Imaging and measurement of embryos and tadpoles**. Tailbud stages were imaged using a Nikon SMZ1500 stereomicroscope coupled to a Nikon DS-Fi1-U2 color camera at ×0.75 magnification with fiber optic illumination using NIS-Elements software. Tadpoles were imaged by placing them in a petri dish filled with water. Images were analysed and length measured head to tail. Length measurements were taken using the line tool in ImageJ.

Embryo movies were acquired with a time-frame of 3 min covering an overall period of 8 h of development, and then displayed at a rate of seven frames per second (fps).

**Cell cycle duration measurement**. Images were captured every 3 min on an Olympus SZX16 stereomicroscope equipped with a Leica DFC450C color camera at ×0.7 magnification with fiber optic illumination using MetaMorph7.8 software. Time-lapse movies of injected embryos were played and divisions were counted to determine the frame number of the forth cleavage. The inter-cleavage period was determined by tracking individual cells and noting the frame number at which the cleavage furrow visibly transected the entire cell. When daughter cells did not divide concurrently, the division time of the earliest dividing daughter was used, and that cell was followed for the remaining time of the movie. When the cleavage could not be observed, as in cases where the cleavage plane did not intersect with the embryo surface, the cell was omitted from analysis.

**Quantitative real-time PCR (qRT-PCR)**. For each condition, four embryos were lysated using TRIzol reagent (Invitrogen), then total RNA was extracted using Direct-zol™ DNA/RNA MiniPrep kit (Zymo Research). First-Strand cDNA synthesis was performed with the iScript™ Advanced cDNA Synthesis Kit (BioRad). qPCR was performed using SsoFast EvaGreen Supermixes (BioRad) and a Roche LightCycler 96. All reactions were performed in triplicate, at a minimum. The

following program was used: 95 °C for 600 s; 45 cycles of 95 °C for 10 s, 60 °C for 30 s; increasing at 0.2 °C/s from 65 °C to 97 °C. Gene expression was normalized to H4 and calculated by the DDCt method. Primer sequences are provided here:

BMP4-Fwd 5′- TCCTGCTCGGAGGCACTAAC -3′,
BMP4-Rev 5′-ACTTTCTTCTTGCCCGTGTCA-3′,
MyoD-Fwd 5′- CAACCAAAGGCTCCCCAAA-3,
MyoD_Rev 5′-GAGGCTCTCTATGTAGCGAATCG -3′,
xVent1-Fwd 5′- CCCAACAAATAAGCAAACTGGAA -3′,
xVent1-Rev 5′- CAGGTGCCCCCAGATATCTC -3′,
xVent2-For 5′- CCAGAACCGCAGGATGAAAT -3′;
xVent2-Rev 5′- GGTATGAGTCTGGTCTGCCATCT -3′,
Gsc-Fwd 5′- AGGCACAGGACCATCTTCACCG -3′,
Gsc-Rev 5′- CACTTTTAACCTCTTCGTCCGC -3′,
Mix1-Fwd 5′- TCAGCCATTTGCCATGAATC -3′,
Mix1-Rev 5′- TGGGATGCTGCTGGAAGTC -3′,
H4-Fwd 5′- AGGGACAACATCCAGGGCATCACC -3′,
H4-Rev 5′- ATCCATGGCGGTAACGGTCTTCCT -3′

**Quantification and statistical analysis**. Statistical analysis (Student's *t*-test, two-tailed) and analysis of variation (ANOVA) were performed with GraphPad Prism 7 software and indicated in figure legends. Images shown represent typical results. All experiments have been repeated at least three times.

**Reporting summary**. Further information on research design is available in the Nature Research Reporting Summary linked to this article.

## Data availability

Data supporting the findings of this manuscript have all been included. A separate source data file contains raw data underlying Figs. 1a, b, d, e, 2a–g, 3a–d, 4a, c, 5a–f, 6b, c, e, f, h, i, 7b, c, e, f, h, i and Supplementary Figs. 2a, c, e, 3a, b, 4, 5a, c, 6, 7 and 8c–e. A reporting summary for this article is available as a Supplementary Information file linked to this manuscript. Proteomics data have been deposited to the ProteomeXchange Consortium via PRIDE with the dataset identifier PXD017383.

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

## Acknowledgements

We thank H. Mahbubani for technical support with *Xenopus laevis*, M. Skehel, P. Soffientini and Angela Bachi for the mass spectrometry analysis and J. Gannon for support with protein biochemistry. This work was funded by Cancer Research UK, the Associazione Italiana per la Ricerca sul Cancro (AIRC), Worldwide Cancer Research (WWCR), the European Research Council (ERC) grant 614541 and the Giovanni-Armenise foundation career development award to V. Costanzo. L.F. was funded by a fellowship from AIRC, CODITAL Rif: 19444. E.R. was funded by a fellowship from AIRC, Love Design Rif:18196.

## Author contributions

L.F., E.R., F.R., F.P., and S.F. performed all the experiments with *Xenopus laevis* egg extract and embryos and produced critical reagents. F.C, I.C., and D.P. performed embryo imaging. L.F. and E.R. analyzed the data. V.C. designed the experiments, analyzed the data and wrote the manuscript.

## Competing interests

The authors declare no competing interests.
