## [Peer Review File · Nature Communications]

Reviewers' comments:

Reviewer #1 (Remarks to the Author):

In this manuscript, the authors try to understand why somatic nuclei preincubated in mitotic *Xenopus* egg extracts are reprogrammed for rapid DNA replication when further incubated in interphase egg extracts. They show that SSRP, by removing histone H1 from somatic nuclei, is the main factor responsible for this regulation. They also show that an excess of SSRP accelerates DNA replication in post-MBT embryos.

This study is rather interesting and the first part of the manuscript is quite convincing. The results are quite significant, both in the field of reprogramming, in the field of development, and in the field of chromatin. However, the second part of the manuscript should be improved to be convincing enough for its publication.

Major comments:

The findings that SSRP1 is responsible for the replication reprogramming of somatic nuclei and that the involved involved is the removal of histone H1 from chromatin are both very important results. However, because an excess of SSRP accelerates the cell cycle post MBT, the reason logically could be a decrease of histone H1 from post-MBT embryos. Western blot measurements of Histone H1 and other histones post-MBT should be performed in order to evaluate whether H1 is deficient relative to the other histones when an excess of SSRP1 is injected.

In Figure 6, the authors should show the amount of SSRP in the extracts in order to control the amount of injected SSRP remaining in the embryo.

Fig 6b, e, and h are difficult to read. They should be improved, or the graphical presentation of the data changed.

In a number of experiments, the quantification of SSRP present in control and injected embryos is not shown. This should have been an important control in such experiments.

The delay in MBT after adding SSRP is barely detectable both in Fig 6b, Fig 6c, and S4b.

A major event at the MBT is the onset of transcription. The authors could include data showing the profile of new transcription in their injection experiments, as done by co-injection of ³²P rNTP in pioneering articles by Newport and Kirshner. Such experiments should be quite easy for this laboratory.

Minor comments

Fig. 1B is not clearly described. Why 400 nuclei? Did the authors try a dose-response curve which showed saturation at 100%? Why were human proteins used instead of *Xenopus* proteins?

Fig 2: curves should be in color, as it is difficult to read them in B&W.

I do not understand Fig S1C, because SS DNA replication should be already 100% in the extract. So how can a reaction be stimulated which is already 100%?

The description of Fig S6 is misleading: The stimulation of transcription by adding SSRP is mainly post MBT and not at the MBT.

Reviewer #2 (Remarks to the Author):

Falbo et al 2019

The cell cycle is dramatically remodelled during early embryogenesis in parallel with a transition from the expression of the maternal to the zygotic genome. This is accompanied by a wide variety of other cellular changes including the maturation of chromatin. Early embryogenesis also can be considered as a transition from a pluripotent state to a differentiated state and therefore may reveal the barriers to de-differentiation that are inherent to somatic cells.

This is a fascinating study that addresses the importance of FACT dependent deposition of histone H1 during xenopus early development. While mitotic extract strongly supports replication initiation of sperm nuclei in vitro, interphase extract does not. The data presented here suggest that this is due to Histone H1 deposition onto chromatin, which is counteracted by SSRP1 of the FACT complex. They further show that Histone H1 deposition inhibits origin licensing, resulting in increased interorigin distances. As replication initiation rates are critical for controlling total cell cycle length at the MBT, increased SSRP1 levels counteract the slowing of the cell cycle at the MBT. Interestingly these faster cell cycles result in accelerated development and viable tadpoles. If the authors can take into account my comments I think this is an important study that is worthy of publication, which will have implications for a wide-variety of fields from cell cycle to development.

Major point

1. While the effects of SSRP1 in this study are convincing, that it fulfils these H1 functions without Spt16 are less convincing. A simple reason why addition of Spt16+Ssrp1 might not increase the effect on replication is that Ssrp1 is limiting in interphase extracts, but Spt16 is not. In addition the NTD of SSRP1, which binds H1 and stimulates H1 removal from chromatin, still binds to Spt16. An experiment to address whether SSRP1 is acting alone or not in Xenopus extracts would be to perform an SSRP1 and/or Spt16 depletion and add back experiment, of SSRP1, Spt16 and both. Since this type of experiment is standard for the Costanzo lab, I assume this experiment has not been done because it can't be achieved for a technical reason. If this is the case, then I would mention this in the paper. If it can be done, then it is certainly worth doing as it would add a great deal of weight to their argument of the necessity and sufficiency of these proteins.

Minor points

1. The labelling on Figure 2A is difficult to understand. I think it means that the proteins were pre-incubated with SN, before incubation in interphase extract. I think this could be better reflected in the key to this figure.

2. I am sure that this would be taken care of in the copy editing stage, but there are a number of sentences that are not very good English. A couple of examples are the 7th line of the results section, the sentence beginning "However, we noticed..." and 8th line p.5 "MS analysis highlighted...". Please go through the manuscript and make the sentences as clear as possible.

3. SSRP1 is misspelled in the line in the introduction beginning "Critically, we show..."

4. I think this sentence in the abstract is overstated because they don't show that SSRP1 decay really is important. "SSRP1 decays, instead, sets the onset of asynchronous somatic cell cycles at mid-blastula transition (MBT), whereas its increase delays it and, surprisingly, accelerates post-MBT cell cycle speed and embryo development. "

Perhaps simply "Increased levels of SSRP1 accelerates post-MBT cell cycle speed and embryo development. "

5. In Figure 7b, length is misspelled.

Response to reviewers' letter

Reviewer #1 (Remarks to the Author):

In this manuscript, the authors try to understand why somatic nuclei pre incubated in mitotic *Xenopus* egg extracts are reprogrammed for rapid DNA replication when further incubated in interphase egg extracts. They show that SSRP, by removing histone H1 from somatic nuclei, is the main factor responsible for this regulation. They also show that an excess of SSRP accelerates DNA replication in post-MBT embryos.

This study is rather interesting and the first part of the manuscript is quite convincing. The results are quite significant, both in the field of reprogramming, in the field of development, and in the field of chromatin. However, the second part of the manuscript should be improved to be convincing enough for its publication.

Our response: We thank this referee for considering our results important for several fields, including reprogramming, development and chromatin biology. We are grateful for the insightful comments, which helped us to further improve our manuscript. We have addressed all the points raised as explained below in details.

Major comments:

The findings that SSRP1 is responsible for the replication reprogramming of somatic nuclei and that the involved involved is the removal of histone H1 from chromatin are both very important results. However, because an excess of SSRP accelerates the cell cycle post MBT, the reason logically could be a decrease of histone H1 from post-MBT embryos. Western blot measurements of Histone H1 and other histones post-MBT should be performed in order to evaluate whether H1 is deficient relative to the other histones when an excess of SSRP1 is injected.

Our answer: We have addressed this legitimate concern by quantifying the levels of histone H1 relative to other histones in post-MBT embryos and included this new data in the manuscript in Supplementary Fig 5a. We now show that total histone H1 levels are not affected by an excess of injected SSRP1 in post-MBT embryos. Therefore, acceleration of post-MBT development cannot be due to a general decrease in total Histone H1 levels.

In Figure 6, the authors should show the amount of SSRP in the extracts in order to control the amount of injected SSRP remaining in the embryo.

Our answer: As requested we have now included immunoblots of injected Myc-SSRP1, Δ NTD and NTD proteins for all the pre- and post- MBT points considered in Figure 6c, 6f and 6i. These new immunoblots clearly show that the injected proteins are expressed in the embryo and remain for the entire duration of the experiment.

Fig 6b, e, and h are difficult to read. They should be improved, or the graphical presentation of the data changed.

Our answer: We have now improved the graphical presentation of figures 6b, 6e and 6h. Importantly, we have also included new measurements relative to a higher number embryos as explained below in more details.

In a number of experiments, the quantification of SSRP present in control and injected embryos is not shown. This should have been an important control in such experiments.

Our answer: As requested we have now included immunoblots showing endogenous SSRP1 protein for control and injected embryos in all the conditions considered in Figure 6c, 6f and 6i. These immunoblots demonstrate that endogenous SSRP1 protein levels are not affected by exogenous injected Myc-SSRP1, Δ NTD and NTD mRNAs.

The delay in MBT after adding SSRP is barely detectable both in Fig 6b, Fig 6c, and S4b.

Our answer: We have now scored more developing embryos in the improved Fig 6b, 6c and Supplementary Fig 5c (previous S4b). From the new panel 6b it is clear that all SSRP1 injected embryos delay the timing of the MBT onset as they perform one more synchronous cycle (15 instead of 14 synchronous cycles of the buffer injected embryos). As previously shown, similar effects can be observed with NTD injection (Fig 6h) but not with Δ NTD mutant protein (Fig 6e).

As far as panel 6c is concerned we have now included CDK1 phospho-Tyrosine 15 immunoblots at low and high exposures, which allow a better appreciation of MBT delay induced by SSRP1 and NTD injections. These data show that phosphorylation of CDK1, which is a sensitive marker of MBT onset, is delayed at all points in SSRP1- and NTD-, but not in Δ NTD-injected embryos. Please, note that in Fig 6c embryos at MBT (Stage 8), have been sampled every 30 min, as now detailed in the figure key. The figure legend associated to this experiment has been improved to better clarify the description of the experiment.

As far as panel S4b (now Supplementary Fig 5c) is concerned we have also included more embryos and improved graphical presentation, which now shows

more clearly that SSRP1 induced delay of MBT can be suppressed by co-injecting an excess of exogenous histone H1 mRNA.

A major event at the MBT is the onset of transcription. The authors could include data showing the profile of new transcription in their injection experiments, as done by co-injection of ³²P rNTP in pioneering articles by Newport and Kirshner. Such experiments should be quite easy for this laboratory.

Our answer: *We thank this referee for this comment, which allowed us to directly show that de novo transcription takes place at MBT. To this end we have measured RNA transcription in control and SSRP1- injected embryos by monitoring de novo RNA transcription with UTP analogue Ethynyl-UTP (EU). EU is incorporated in total nascent RNA produced in nucleo-plasmic extracts made from stage 9 post MBT embryos injected with buffer or SSRP1 and can be detected and quantified by monitoring fluorescent labelling of extracted RNA (Supplementary Fig 8a and 8d-e). The results of this new experiment show that SSRP1 stimulates RNA transcription. These new results are in agreement with the qPCR data already presented in Supplementary Fig 7.*

Minor comments

Fig. 1B is not clearly described. Why 400 nuclei? Did the authors try a dose-response curve which showed saturation at 100%? Why were human proteins used instead of Xenopus proteins?

Our answer: *In Fig 1B we now show the nuclei titration that led us to deduce that somatic nuclei were not replicated as efficiently as sperm nuclei when similar amount of DNA is incubated in egg extract. We choose 400 nuclei/microliter as starting point of the dose-response experiment for technical reasons as this was the minimum amount of DNA that could be easily detected in a typical replication assay and we increased it 5- and 10- fold, respectively. We have also specified that the graph shows the amount of replicated DNA compared to its corresponding input. Replicated DNA is rarely 100% due to intrinsic quality of egg extracts. From this experiment, it is clear that in interphase egg extract 4000 somatic nuclei/ μ l are not replicated as efficiently as 400 somatic nuclei/ μ l or as efficiently as an equivalent amount of sperm DNA, which corresponds to 8000 aploid sperm nuclei/microliter. This is the main information that we could infer from this experiment, which allowed us to formulate the hypothesis that a limiting factor prevents full replication of high amounts of somatic nuclei when incubated directly in in interphase egg extract. Human proteins were used as, in contrast to the Xenopus ones, they could be expressed in soluble form at the high amounts required for the experiments shown, as we now mention in the manuscript text. Given the high similarity of conserved chromatin proteins, including histones and histone chaperones, human and Xenopus versions can be used interchangeably.*

Fig 2: curves should be in color, as it is difficult to read them in B&W.

Our answer: *We have now included color coding for curves in Fig 2 as requested.*

I do not understand Fig S1C, because SS DNA replication should be already 100% in the extract. So how can a reaction be stimulated which is already 100%?

Our answer: *We agree with this referee that ssDNA is replicated with maximum efficiency in egg extract. Therefore, we have removed this experiment, which is not highly informative.*

The description of Fig S6 is misleading: The stimulation of transcription by adding SSRP is mainly post MBT and not at the MBT.

Our answer: *We have now better specified that stimulation starts around MBT and proceeds post-MBT to be more accurate.*

Reviewer #2 (Remarks to the Author):

Falbo et al 2019

The cell cycle is dramatically remodelled during early embryogenesis in parallel with a transition from the expression of the maternal to the zygotic genome. This is accompanied by a wide variety of other cellular changes including the maturation of chromatin. Early embryogenesis also can be considered as a transition from a pluripotent state to a differentiated state and therefore may reveal the barriers to de-differentiation that are inherent to somatic cells. This is a fascinating study that addresses the importance of FACT dependent deposition of histone H1 during xenopus early development. While mitotic extract strongly supports replication initiation of sperm nuclei in vitro, interphase extract does not. The data presented here suggest that this is due to Histone H1 deposition onto chromatin, which is counteracted by SSRP1 of the FACT complex. They further show that Histone H1 deposition inhibits origin licensing, resulting in increased interorigin distances. As replication initiation rates are critical for controlling total cell cycle length at the MBT, increased SSRP1 levels counteract the slowing of the cell cycle at the MBT. Interestingly these faster cell cycles result in accelerated development and viable tadpoles. If the authors can take into account my comments I think this is an important study that is worthy of publication, which will have implications for a wide-variety of fields from cell cycle to development.

Our response: *We thank this referee for the very positive appreciation of our study, which we agree will have implications for a number of different fields. We*

are grateful for the insightful comments, which helped us to clarify some aspects of our findings. Detailed response to the points raised can be found below.

Major point

1. While the effects of SSRP1 in this study are convincing, that it fulfils these H1 functions without Spt16 are less convincing. A simple reason why addition of Spt16+Ssrp1 might not increase the effect on replication is that Ssrp1 is limiting in interphase extracts, but Spt16 is not. In addition, the NTD of SSRP1, which binds H1 and stimulates H1 removal from chromatin, still binds to Spt16. An experiment to address whether SSRP1 is acting alone or not in *Xenopus* extracts would be to perform an SSRP1 and/or Spt16 depletion and add back experiment, of SSRP1, Spt16 and both. Since this type of experiment is standard for the Costanzo lab, I assume this experiment has not been done because it can't be achieved for a technical reason. If this is the case, then I would mention this in the paper. If it can be done, then it is certainly worth doing as it would add a great deal of weight to their argument of the necessity and sufficiency of these proteins.

Our response: *We have attempted the depletion of SPT16 to exclude the possibility that exogenous SSRP1 is acting through endogenous SPT16. However, we could not successfully deplete SPT16 from egg extract. Previous quantification of *Xenopus* proteome from the Kirschner lab has shown that SPT16 is present at a concentration of about 620 nM in egg extract (Wuhr et al Curr Biol 2014). We could not achieve complete removal of SPT16 using antibodies raised in the lab probably due to this high concentration. This was not unexpected as some histone proteins present at a similar high concentration in egg extract, cannot be completely depleted with antibodies. We have now clearly stated that we could not deplete SPT16 from egg extract, as suggested. In order to provide alternative support to our hypothesis that SSRP1 is acting independently of endogenous SPT16 we tested whether SSRP1 is able to bind somatic nuclei, which do not contain detectable levels of SPT16 (Supplementary Fig 3a). To this end we incubated recombinant SSRP1 protein with somatic chromatin in the absence of egg extract. As we now show in Supplementary Fig 3a SSRP1 can bind somatic chromatin in the absence of SPT16. Also, the participation of SPT16 in histone H1 removal from chromatin in the presence of an excess of exogenous SSRP1 protein would likely entail the increased recruitment of SPT16 to chromatin. However, we now show in Supplementary Fig 3b that an excess of exogenous SSRP1 does not stimulate further recruitment of endogenous SPT16 onto chromatin when egg extract is added to somatic nuclei pre-incubated with recombinant SSRP1 protein. Collectively, these new experiments suggest that SSRP1 can act independently of SPT16, although they do not completely rule out a role for SPT16, as we now carefully discuss in the manuscript.*

Minor points

1. The labelling on Figure 2A is difficult to understand. I think it means that the

proteins were pre-incubated with SN, before incubation in interphase extract. I think this could be better reflected in the key to this figure.

Our response: *We have now specified in figure 2A legend that proteins were pre-incubated with somatic nuclei for 30 min before transferring these to extracts.*

2. I am sure that this would be taken care of in the copy editing stage, but there are a number of sentences that are not very good English. A couple of examples are the 7th line of the results section, the sentence beginning “However, we noticed...” and 8th line p.5 “MS analysis highlighted...”. Please go through the manuscript and make the sentences as clear as possible.

Our response: *We have now addressed the sentences highlighted by this referee and improved the manuscript text.*

3. SSRP1 is misspelled in the line in the introduction beginning “Critically, we show...”

Our response: *Thank you for noticing the mistake, which we have now corrected.*

4. I think this sentence in the abstract is overstated because they don't show that SSRP1 decay really is important. “SSRP1 decays, instead, sets the onset of asynchronous somatic cell cycles at mid-blastula transition (MBT), whereas its increase delays it and, surprisingly, accelerates post-MBT cell cycle speed and embryo development. “

Perhaps simply “Increased levels of SSRP1 accelerates post-MBT cell cycle speed and embryo development.

Our answer: *We have now modified the sentence as suggested, removing the causality link between SSRP1 decay and the onset of MBT.*

5. In Figure 7b, length is misspelled.

Our answer: *Thank you, mistake corrected.*

REVIEWERS' COMMENTS:

Reviewer #1 (Remarks to the Author):

In my first report on this manuscript, I was impressed by the novelty and importance of the results. I had only a few concerns about the first part of the manuscript showing that SSRP1, a subunit of the FACT complex, was involved in the removal of histone H1 from chromatin. This result explained an important aspect of the reprogramming of somatic nuclei incubated in *Xenopus laevis* mitotic egg extracts. This part of the revised manuscript is still very convincing.

My main concerns were about the results obtained by microinjection experiments made in *X. laevis* embryos. Several new data are presented in the revised manuscript to address my suggestions. I therefore consider that this manuscript is suitable for publication.

Two small remarks:

In Figure 1a, the legend in the figure itself should be "SN+ Mitosis + Interphase" and not "SN+ Mitosis", which is misleading.

Figure 6b can be quite misleading if the legend is not carefully read. The graph indicates a number of divisions but does not specify that it is over a precise time (330 min). Therefore, it looks like that there is no control at division 15. I recommend a longer explanation in the corresponding main text and panel 6b,e,f (legend is fine) should be clarified.

Reviewer #2 (Remarks to the Author):

The authors have addressed my minor concerns and the paper is in my opinion appropriate for publication.

Point by point response to REVIEWERS' COMMENTS:

Reviewer #1 (Remarks to the Author):

In my first report on this manuscript, I was impressed by the novelty and importance of the results. I had only a few concerns about the first part of the manuscript showing that SSRP1, a subunit of the FACT complex, was involved in the removal of histone H1 from chromatin. This result explained an important aspect of the reprogramming of somatic nuclei incubated in *Xenopus laevis* mitotic egg extracts. This part of the revised manuscript is still very convincing.

My main concerns were about the results obtained by microinjection experiments made in *X. laevis* embryos. Several new data are presented in the revised manuscript to address my suggestions. I therefore consider that this manuscript is suitable for publication.

Our response:

We thank this reviewer for the positive evaluation and the insightful criticisms, which has helped to improve our manuscript.

Two small remarks:

In Figure 1a, the legend in the figure itself should be “SN+ Mitosis + Interphase” and not “SN+ Mitosis”, which is misleading.

Our response:

We have now corrected this point as suggested.

Figure 6b can be quite misleading if the legend is not carefully read. The graph indicates a number of divisions but does not specify that it is over a precise time (330 min). Therefore, it looks like that there is no control at division 15. I recommend a longer explanation in the corresponding main text and panel 6b,e,f (legend is fine) should be clarified.

Our response:

We have now clarified in the text and in the figure legend of Figure 6 that the graphs in the indicated panels show the number of synchronous divisions-only monitored up to 450 minutes from fertilization.

Reviewer #2 (Remarks to the Author):

The authors have addressed my minor concerns and the paper is in my opinion appropriate for publication.

Our response:

We thank this referee for the positive evaluation of our manuscript.